# MoSA: Motion-constrained Stress Adaptation for Mitigating Real-to-Sim Gap in Continuum Dynamics via Learning Residual Anisotropy

Jiaxu Wang[† * 1 2]  Junhao He[* 1]  Jingkai Sun[3]  Yi Gu[1]  Yunyang Mo[1]  Jiahang Cao[3]  Qiang Zhang[1]
Renjing Xu[† 1]

## Abstract

Learning real-world dynamics from visual observations is crucial for various domains. A common strategy is to calibrate simulators by estimating physical parameters, yet accuracy is ultimately bounded by the underlying physical models, which often assume materials are homogeneous and isotropic. Even if reasonable, real-world objects typically exhibit mild anisotropy and heterogeneity. After the near-isotropic backbone is well calibrated, these residual effects become the key bottleneck for further closing the real-to-sim gap. Although neural networks can fit dynamics end-to-end, such black-box modeling discards strong physical priors, leading to poor data efficiency and overfitting. Therefore, we propose MoSA, a motion-constrained stress adaptation framework that targets these residual effects to further improve real-to-sim dynamics learning. MoSA uses an isotropic model as a physics prior and learns residual stress operators to capture mild anisotropy and heterogeneity. It progressively adapts stresses via microplane-constrained redistribution in a physics-informed cascaded network. We further impose motion constraints by supervising temporal and spatial derivatives of the deformation field. Experimentally, our learned dynamics achieves superior accuracy, generalization, and robustness, while learning physically meaningful residual anisotropy. Finally, we validate MoSA in a robot manipulation setting, showing that better real-to-sim dynamics modeling translates into more reliable sim-to-real

transfer. Project Page is available at `https://mercerai.github.io/MoSA/`.

## 1. Introduction

Simulating realistic dynamics of diverse materials is essential for many applications in graphics, robotics, and embodied agents, and it is a key ingredient for building interactive digital twins of the real world. Visual observations are widely available and naturally capture rich deformation and motion signals. As a result, learning real-world dynamics from visual data has become an active research direction.

A prevailing paradigm is to represent dynamics with classical physical models and identify objects by estimating a small set of physical parameters (Cai et al., 2024; Li et al., 2023). In practice, most pipelines adopt homogeneous and isotropic material models, since they provide a simple and often effective approximation for many everyday objects and enable stable differentiable simulation. However, this approximation is rarely exact: even when an object can be reasonably treated as homogeneous and isotropic, it typically exhibits small but systematic anisotropy and heterogeneity in the real world (Li et al., 2025; Fu et al., 2025). These deviations, which we call residual effects in continuum, can arise from internal structure, gradual wear, etc. As a result, estimating parameters within an isotropic model can capture the dominant backbone, but it often fails to account for these residual effects, which become a key bottleneck for further closing the real-to-sim gap. An alternative is to learn dynamics end-to-end with neural networks (Mittal et al., 2025; Cao et al., 2025; Zhobro et al., 2025). While this can model missing physics, directly fitting full dynamics discards strong physical priors and does not disentangle the dominant isotropic backbone from weak residual effects. The network must re-learn the backbone and capture subtle residuals at the same time, which often leads to poor data efficiency, unstable training, and overfitting.

Motivated by this bottleneck, we take a targeted modeling approach: we keep a calibrated near-isotropic backbone to explain the dominant behavior, and learn only the residual effects that arise from mild anisotropy and heterogene-

*Equal contribution  [1]Hong Kong University of Science and Technology, Guangzhou, China [2]MMLab, Chinese University of Hong Kong, Hong Kong SAR [3]The University of Hong Kong, Hong Kong SAR. Correspondence to: Jiaxu Wang <jiaxuwang@cuhk.edu.hk>, Renjing Xu <renjingxu@hkust-gz.edu.cn>.

*Proceedings of the 43rd International Conference on Machine Learning*, Seoul, South Korea. PMLR 306, 2026. Copyright 2026 by the author(s).

ity. Concretely, we treat an isotropic constitutive law as a physics prior and introduce a structured residual stress adaptation operator that progressively corrects the prior stress response and encourage physically consistent stress redistribution. This design preserves strong physical inductive bias while allocating learning capacity to the missing residual dynamics.

Another challenge in learning dynamics from vision is that supervision is often indirect and underconstrained. Earlier attempts assume access to full 3D geometry trajectories (Ma et al., 2023; Jaques et al., 2019), which is hard to obtain in real-world settings. More recent methods (Chen et al., 2025a) connect videos to 3D simulation through differentiable simulation (Bolliger et al., 2025) and neural rendering (Kerbl et al., 2023; Wang et al., 2024b). However, optimization driven only by image reconstruction remains ill-conditioned: different 3D deformation trajectories and material responses can produce similar image appearances.

To mitigate this, we add motion-aware constraints that provide more direct supervision than pixel reconstruction alone. We first obtain a dynamic 3D reconstruction from the input videos, which provides reliable cues about how the object moves and deforms over time. We then use these cues to supervise the temporal and spatial derivatives of the deformation field. Together with our targeted residual stress adaptation operator, this yields our framework, MoSA.

- We propose a physics-informed residual stress adaptation module, which augments an isotropic constitutive prior with bounded residual operators to capture the mild anisotropy and heterogeneity commonly present in real-world objects. By explicitly correcting the prior stress response in a structured and progressive manner, it improves simulation fidelity while preserving physical inductive bias and partial interpretability.
- We introduce a motion-constrained optimization strategy that leverages motion cues from dynamic reconstruction to supervise temporal and spatial derivatives of the deformation field. This higher-order supervision provides more direct constraints than pixel reconstruction alone, improving data efficiency and reducing overfitting in video-based dynamics learning.
- Experiments on both synthetic and real data demonstrate superior accuracy, generalization, and robustness, and analyses confirm that our model learns physically meaningful residual anisotropy. We further validate the practical impact of improved real-to-sim dynamics in a robot manipulation setting.

## 2. Related Works

**Neural Physical Dynamics**. Many prior works use neural networks to model 3D dynamics, often predicting fu-

ture states from current geometry (Jing et al., 2026; Xue et al., 2024; Mittal et al., 2025; Zhobro et al., 2025; Chen et al., 2025c). Graph-based simulators (Wei & Freris, 2025; Zhang et al., 2024) employ GNNs to achieve plausible results across materials, but can overfit due to their high parameter count. To mitigate this, some methods inject physical structure by embedding neural components into PDEs; for instance, (Mittal et al., 2025) learns a unified latent constitutive law to represent material types. NCLaw (Ma et al., 2023) replaces constitutive laws with neural networks. DEL (Wang et al., 2024a) extends the Discrete Element Method with neural operators to model diverse materials. MASIV (Zhao et al., 2025) supervises particle trajectories with isotropic neural dynamics, while our method uses higher-order motion constraints and learns anisotropic residual stress corrections. OmniPhysGS (Lin et al., 2025) adopts discrete isotropic experts for plausible dynamics, whereas our method models continuous heterogeneity and anisotropic residuals for accurate real-to-sim dynamics. Recent efforts such as (Cao et al., 2025; Chen et al., 2025b) further emphasize incorporating physical priors to improve generalization. Despite this progress, many neural dynamics models effectively learn full dynamics or full operators. This often underuses strong explicit priors and does not explicitly separate the dominant near-isotropic behavior from the weaker residual effects that matter most in real-to-sim settings, which can hurt data efficiency and generalization. In contrast, we keep an isotropic constitutive prior as the backbone. We then learn residual stress corrections that target mild anisotropy and heterogeneity to breakup the bottleneck.

**System Identification**. Earlier works have explored estimating physical parameters directly from visual input (Liu et al., 2025; Chen et al., 2024), aided by advances in differentiable physics simulation . These methods (Jiang et al., 2025; Vasile et al., 2025; Xu et al., 2025) often compare rendered outputs with 2D ground truth, achieving promising results but are generally limited to elastic or rigid materials and can suffer from reconstruction artifacts. PAC-NeRF (Li et al., 2023) jointly optimizes geometry and physical parameters from multi-view videos but is hindered by rendering and geometry errors. Spring-GS (Zhong et al., 2024) uses a spring-mass model for simulatable elastic objects, while GIC (Cai et al., 2024) improves physical estimation using shape constraints from dynamic 3DGS. Vid2Sim (Chen et al., 2025a) adopts a ViT for parameter estimation with a Simplicits (Modi et al., 2024) for refinement. These methods estimate parameters with a fixed physical model family, so accuracy is bounded by model misspecification. Our approach retains the isotropic backbone and learns residual dynamics beyond it.

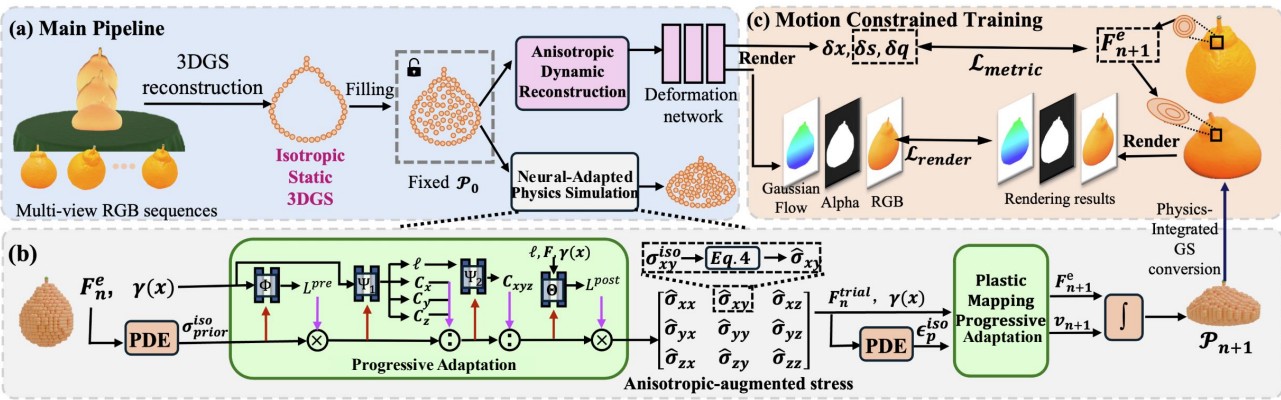

*Figure 1.* Overview of the pipeline. (a) Two-stage dynamic reconstruction. (b)Simulation with progressive anisotropic stress adaptation (c) Motion-constrained optimization strategy

## 3. Methodology

**Problem Definition**. This work aims to learn real-to-sim dynamics from multi-view videos by preserving an isotropic constitutive prior as the backbone and modeling the residual effects induced by mild anisotropy and heterogeneity. Given multi-view video sequences $V_i \mid i = 1, 2, \ldots, n$ of a deforming object with known camera parameters, our goal is to estimate the global physical parameters of the isotropic prior and learn residual stress correction operators that compensate for the mismatch between simplified constitutive assumptions and real materials. The overall pipeline is illustrated in Fig. 1. Notably, we do not aim to model strongly anisotropic materials from scratch; instead, we target mild continuum residuals that often account for the remaining real-to-sim error when the common isotropic pipelines are adopted to model real-world dynamics.

**Dynamic Gaussian Splatting**. 3DGS represents scenes explicitly by rendering a set of Gaussians using efficient differentiable rasterization. Each Gaussian is defined by its mean $x_0$, covariance $\Sigma$, opacity $o$, and color via spherical harmonics, with the form $G(x) = e^{-\frac{1}{2}(x-x_0)^T \Sigma^{-1}(x-x_0)}$. To ensure $\Sigma$ is positive semi-definite, it is decomposed as $\Sigma = RSS^T R^T$, where $R$ is a rotation matrix and $S$ a scale matrix. During rendering, the color and mask are computed by blending $N$ ordered Gaussians overlapping each pixel.

$$I(u) = \sum_{i \in N} T_i \alpha_i c_i, \quad A(u) = \sum_{i \in N} T_i \alpha_i, \quad (1)$$

where $T_i = \prod_{j=1}^{i-1}(1 - \alpha_j)$. $\alpha_i(x)$ is the projected 2D occupancy $\alpha_i(x) = o_i exp(-\frac{1}{2}(x - \mu_i)^T \Sigma_i^T(x - \mu_i))$.

The common paradigm of dynamic 3DGS is to use a deformation model to map static Gaussian splats to other timestamps, as in:

$$(\delta \mathbf{x}, \delta \mathbf{s}, \delta \mathbf{q}) = \mathcal{F}_\theta(\gamma(\mathbf{x}), \gamma(\mathbf{t})). \quad (2)$$

Different methods define $\mathcal{F}_\theta$ differently, e.g., an MLP in (Yang et al., 2024), a neural triplane in (Wu et al., 2024),

and a set of motion primitives in GIC (Cai et al., 2024). For fair comparisons, we adopt the method from GIC as our reconstruction tool.

**Constitutive Model**. Similar to prior works, we represent object geometry as particles and simulate using the Material Point Method (MPM) (Jiang et al., 2016). A core component of MPM is the constitutive model, which defines how objects deform under external forces by relating strain ($\epsilon$) to stress ($\sigma$), typically as $\sigma = \boldsymbol{\sigma}(\epsilon)$, where $\epsilon$ is derived from the deformation gradient $\mathbf{F}$ (e.g., Green strain: $\epsilon = \frac{1}{2}(\mathbf{F}^T \mathbf{F} - I)$). In elastoplastic mechanics, constitutive models include an elastic response and a plastic projection that maps excessive strain back to admissible configurations, expressed as $\hat{\epsilon} = \boldsymbol{\epsilon}(\epsilon)$ or $\hat{F} = \boldsymbol{\epsilon}(F)$. More details are provided in the **Appendix P**.

### 3.1. A Structured, Progressive Stress Adaptation for Modeling Anisotropic Residual Effect

**Problem Analysis about Anisotropic Modeling**. Classic constitutive models often assume materials are homogeneous and isotropic. Under these assumptions, mechanical behavior can be described by only a few parameters (e.g., Young's modulus and Poisson's ratio), which greatly simplifies modeling and enables stable calibration. However, this approximation is rarely exact in the real world. Even when an object is reasonably treated as near-isotropic at a coarse level, it typically exhibits mild anisotropy and spatial heterogeneity due to internal structure, fabrication, or wear. After the near-isotropic backbone is calibrated, these residual effects can dominate the remaining real-to-sim error. Among them, anisotropy is generally harder to express and identify than heterogeneity, since it depends on directional responses rather than only spatial variation. Our method therefore preserves an isotropic constitutive prior as the backbone and learns structured residual corrections for both effects, with particular focus on residual anisotropy as the more challenging component.

Here we take the simple linear anisotropy as an example. Let the strain at a material point be $\epsilon_{kl}$, a symmetric second-order tensor for $k, l = 1, 2, 3$. In conventional linear isotropic models, the stress is computed as $\sigma_{kl} = \mathbf{C}_{kl}\epsilon_{kl}$, where $\mathbf{C}_{kl}$ is a $3 \times 3$ stiffness matrix defined by just two independent parameters, Young's modulus and Poisson's ratio, yielding 2 degrees of freedom. In contrast, linear anisotropic models use tensor contraction: $\sigma_{ij} = \mathbf{C}_{ijkl}\epsilon_{kl}$, where $\mathbf{C}_{ijkl}$ is a $3 \times 3 \times 3 \times 3$ fourth-order stiffness tensor. Accounting for symmetry properties ($\mathbf{C}_{ijkl} = \mathbf{C}_{jikl}$ and $\mathbf{C}_{ijkl} = \mathbf{C}_{ijlk}$), this tensor still has 36 independent parameters, making it far more expressive—but also much more complex to model. In other words, the anisotropic stiffness tensor has 36 degrees of freedom, much higher than the isotropic case, making manual specification impractical. And this is only for linear anisotropy; the complexity increases further in nonlinear settings.

**Anisotropic-informed Architecture Design**. To address the challenges outlined above, we do not formulate a physical equation entirely from scratch. Instead, we correct and refine a prior isotropic constitutive model to capture anisotropy and heterogeneity. Specifically, we design a physics-informed network as a projection function that maps isotropic stress to anisotropic configurations based on the current material state. The projection is defined as $\sigma_{ij} = f_\theta(\mathbf{C}_{kl}\epsilon_{kl}|s)$ in which $s$ denotes the current state (including deformation gradient, prior stress, position, etc.). Moreover, rather than modeling the function $f$ directly with a full black-box network, we define it in a specific form:

$$\hat{\sigma}_{ij} = T_\sigma^{-1} \sum_{kl}(I + \mathbb{C}_{ijkl}^\Psi)T_\sigma \cdot (\boldsymbol{\sigma}(\epsilon)_{kl} + L_{kl}^{\Phi,pre}) + L_{ij}^{\Theta,post}, \quad (3)$$

where $\boldsymbol{\sigma}(\epsilon)_{kl}$ is the prior isotropic constitutive model, $I$ is an identity matrix, $\hat{\sigma}_{ij}$ is the corrected stress incorporating anisotropy. The $T_\sigma$ is defined to map stress tensors from the global coordinate system to the material (or object) coordinate system. This is essential because the constitutive relations for anisotropic materials are inherently defined along their principal directions. The correction consists of three learnable terms: the redistribution term $\mathbb{C}_{ijkl}^\Psi$, the pre-linear term $L_{kl}^{\Phi,pre}$, and the post-linear term $L_{ij}^{\Theta,post}$, each predicted by a cascaded neural network with parameters $\Psi$, $\Phi$, and $\Theta$, respectively. These corrections are applied progressively to adjust the prior stress. This formulation can be interpreted as first correcting the isotropic prior stress by $L_{kl}^{\Phi,pre}$ and then rescaling and redistributing it along different directions by $\mathbb{C}_{ijkl}^\Psi$ and $L_{ij}^{\Theta,post}$. Details are illustrated in the lower panel of Fig. 1 and the following section.

First, $\mathbb{C}_{ijkl}^\Psi$ is a fourth-order tensor, and we decompose the tensor contraction into four matrix multiplications for learning efficiency. Further details will be provided later. The architecture of the cascaded network can be seen in the lower panel of Fig. 1. $L^{\Phi,pre}$ is defined as an isotropic

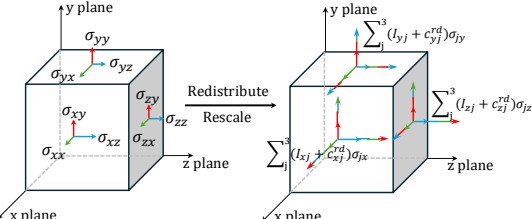

*Figure 2.* Rescaling and redistributing of stress tensor

complementary matrix since we assume that the prior constitutive equation, even when describing isotropy, still has some deficiencies. Therefore, it is expected to make slight adjustments to the isotropic stress. We produce this term by:

$$\mathbf{L}^{pre} = W^{pre}\Phi(U, \sigma_p) + b^{pre}, \quad (4)$$

where $\Phi$ is an MLP with two hidden layers. $\sigma_p$ refers to the prior stress $\sigma_p = \boldsymbol{\sigma}(\epsilon)$. $U$ is the Right Polar Tensor of the deformation gradient obtained by Singular Value Decomposition ($\mathbf{F} = \mathbf{R}\mathbf{U}$). $W^{pre}$ and $b^{pre}$ transform latent vectors into the stress domain.

Second, $\mathbb{C}$ is the fourth-order tensor for contracting with the isotropic stress, which plays the most important role in this pipeline. Unlike the previously stated fixed elastic stiffness tensor, $\mathbb{C}$ is an adaptive correction term dynamically predicted by the network based on the current state of each material point. Directly predicting a full fourth-order tensor is difficult and can lead to numerical instability. To address this, we decompose $\mathbb{C}$ into four second-order tensors $\mathbf{C}_x$, $\mathbf{C}_y$, $\mathbf{C}_z$, and $\mathbf{C}_{xyz}$ with a microplane-based stress redistribution constraint (Bažant Z P, 1996), and apply them sequentially to stress correction. We provide the detailed derivation of the microplane-based constrains in **Appendix** O. This design improves learning stability and efficiency, while providing better physical grounding. Specifically, we rescale and redistribute each component of the stress element, as shown in Fig. 2, considering only those lying on the same microplane during adjustment. That is, each corrected stress is a linear combination of all stress components on its corresponding microplane. For clarity, we use the x-plane as an example in the following.

$$[\hat{\sigma}_{xx}, \hat{\sigma}_{xy}, \hat{\sigma}_{xz}]^T = (\mathbf{I} + \mathbf{C}_x)[\sigma_{xx}, \sigma_{xy}, \sigma_{xz}]^T \quad (5)$$

where $\mathbf{I}$ is an identity matrix, $\mathbf{C}_x$ is a redistribution coefficient matrix associated with the x-plane. When $\mathbf{C}$ is set to zero, the stress degrades to the prior stress. This allows the network to focus only on learning the residual values. If we expand the $\hat{\sigma}_{xx}$ in Eq. 5, we can obtain:

$$\sum_j^{x,y,z}(I_{xj}+c_{xj})\sigma_{xj} = (1+c_{xx})\sigma_{xx}+c_{xy}\sigma_{xy}+c_{xz}\sigma_{xz} \quad (6)$$

The first term serves as the rescaling while the subsequent terms handle the redistribution.

Similarly, we apply $\mathbf{C}_y$ and $\mathbf{C}_z$ to the y- and z-planes. Next, the *mutual equivalence of shear stresses theorem* ($\sigma_{yx} = \sigma_{xy}, \sigma_{zx} = \sigma_{xz}$) is considered to merge paired shear stress components, i.e. $\hat{\sigma}_{xy} = 0.5(\hat{\sigma}_{xy} + \hat{\sigma}_{yx})$. We then reuse Eq. 5 with another redistribution tensor $\mathbf{C}_{xyz}$ to further redistribute the updated normal stress components $\hat{\sigma}_{xx}, \hat{\sigma}_{yy}, \hat{\sigma}_{zz}$, that is $(\mathbf{I} + \mathbf{C}_{xyz})[\sigma_{xx}, \sigma_{yy}, \sigma_{zz}]^T$. This step enhances the modeling of anisotropic effects along the three principal directions. Since the refinement at each microplane occurs at the same time, we use a network to jointly predict $\mathbf{C}_x, \mathbf{C}_y$ and $\mathbf{C}_z$ (Eq. 7).

$$
\begin{aligned}
l_1 &= \Psi_1(U, \sigma_p + \mathbf{L}^{pre}), \\
\mathbf{C}_x, \mathbf{C}_y, \mathbf{C}_z &= \alpha_1 \cdot \tanh(W^c l_1 + b^c), \\
\mathbf{C}_{xyz}, l_2 &= \alpha_2 \cdot \tanh(\Psi_2([l_1, \hat{\sigma}_{xx}, \hat{\sigma}_{yy}, \hat{\sigma}_{zz}])).
\end{aligned} \tag{7}
$$

$\Psi_1$ and $\Psi_2$ are two MLPs. $\alpha_1$ and $\alpha_2$ are two empirical coefficients that govern the effect of the prior stress; the larger $\alpha$, the more deviation from the prior stress is allowed. We constantly set the two $\alpha$s as 0.1; the reason is presented in the **Appendix** D. Applying these $\mathbf{C}$s separately is equivalent to performing a tensor contraction with $\mathbb{C}$, which we derive in the **Appendix** O. Similar to $\mathbf{L}^{pre}$, $\mathbf{L}^{\Theta,post}$ is another linear adjustment applied after the $\mathbb{C}$ is contracted.

$$
\mathbf{L}^{post} = W^{post}\Theta(l_2, U, \hat{\boldsymbol{\sigma}}) + b^{post}. \tag{8}
$$

in which $l$ is the latent code from Eq. 7. $\hat{\boldsymbol{\sigma}}$ is the corrected stress from the previous step. Since the stress tensor should be symmetric, both $\mathbf{L}^{pre}$ and $\mathbf{L}^{post}$ are symmetrized as $\mathbf{L} = \frac{1}{2}(\mathbf{L} + \mathbf{L}^T)$. To ensure all $\mathbf{C}$ matrices are invertible, we predict an upper triangular matrix $\mathbf{U}_c$ and a lower triangular matrix $\mathbf{L}_c$ for each $\mathbf{C}$, and construct $\mathbf{C}$ by multiplying them. To guarantee invertibility, we add a small diagonal offset to both $\mathbf{L}_c$ and $\mathbf{U}_c$.

By the same logic, this paradigm can also be applied to correct the plastic yield projection.

$$
\hat{\Lambda}_{ij} = T_\sigma^{-1} \sum_{kl}(I + \mathbb{C}_{ij}^\Psi)T_\sigma(\Sigma(\tilde{\epsilon}) + L_j^{\Phi,pre}) + L_i^{\Theta,post}, \tag{9}
$$

where $\Sigma(\tilde{\epsilon})$ represents the singular value vector of the deviatoric principle strain tensor. The corrected version can be interpreted as an *equivalent deviatoric principal strain* tensor that conforms to the new anisotropic plasticity criterion.

**Heterogeneity Modeling**. To model material heterogeneity, instead of assigning an independent set of material parameters to each particle (as was done in previous works (Dagli et al., 2025; Lin et al., 2025)), we estimate a single global material parameter that characterizes the overall behavior of the object, together with a continuous implicit field that provides local variations of the global parameter at each spatial location within the object. The physical parameters for position $\mathbf{x}$ are defined as $\mathbf{P}(\mathbf{x}) = \mathbf{P}_{\text{global}} \cdot (1 + 0.2 \cdot tanh(\eta(\mathbf{x})))$.

This approach yields a smoother and more physically consistent representation, avoiding the discontinuities and abrupt jumps that may arise from particle-wise parameterization. To further stabilize the learning process, we apply a $0.2 \cdot tanh(\cdot)$ activation on the network output, constraining it to the range $[-0.2, 0.2]$, and encourage the network predictions to have zero mean and minimal variance. This design can not only capture the global material parameter while allowing controlled local variations.

### 3.2. Motion-constrained Optimization Strategy

Most prior works supervise dynamics learning using rendering losses that align rendered images and masks with the input videos. Such losses provide only indirect and coarse constraints on the deformation field, mainly through silhouettes and boundaries. Since image formation is many-to-one, different 3D deformations and material responses can lead to similar rendered appearances, making the optimization ill-conditioned. This motivates introducing additional motion-aware constraints that provide more direct supervision on the underlying deformation.

Dynamic 3D sequences reconstructed from multi-view videos provide informative motion cues for learning object dynamics. We leverage these cues to better constrain dynamics optimization. Following (Cai et al., 2024), we reconstruct dynamic scenes from multi-view videos with dynamic 3DGS. To ensure accurate initial geometry and improve the capture of local deformations, we adopt a two-stage reconstruction procedure that decouples static geometry from dynamic deformation.

We first reconstruct a static 3DGS at the initial timestamp and apply the filling strategy in (Cai et al., 2024) to obtain the initial particle representation at $t = 0$, denoted as $\mathcal{P}_0$. At this stage, we model Gaussian primitives as isotropic, with equal scales and identity rotations, which provides a clean and stable initialization of the geometry. We then fix $\mathcal{P}_0$ as the point scaffold and train a dynamic 3DGS on top of it. During dynamic training, we allow the Gaussian primitives to have anisotropic covariances, so that they can deform and rotate freely over time through $\mathcal{F}_\theta$ (Eq. 2). This decoupling prevents the dynamic model from absorbing static geometry errors and encourages $\mathcal{F}_\theta$ to focus on learning motion and deformation. Therefore, we further introduce higher-order supervision that directly constrains the temporal and spatial derivatives of the deformation field using motion cues from the dynamic reconstruction.

We begin with the analysis in (Xie et al., 2024) to bridge the deformation of Gaussian splats to the time-varying deformation gradients in simulation. if a Taylor series expansion of the deformation field $\tilde{\phi}_p$ is performed, $\tilde{\phi}_p(X, t) = x_p + F_p(X - X_p)$ can be obtained, in which $F_p$ is the deformation gradient at particle $p$. Substituting the formula into Gaus-

sians: $G_p(\mathbf{x}) = e^{-\frac{1}{2}\left(\phi^{-1}(x,t)-X_p\right)^T \mathbf{A}_p^{-1}\left(\phi^{-1}(x,t)-X_p\right)}$, a $F$-dependent covariance matrix $A_{p,t} = F_{p,t}A_{p,0}(F_{p,t})^T$ is obtained. Combining all into the tensor format can yield:

$$\mathbf{A}_t = \mathbf{F}_t\mathbf{A}_0\mathbf{F}_t^T. \qquad (10)$$

This formulation allows 3DGS rendering to be expressed as a function of both simulated particle positions and their corresponding deformation gradients. This forms the foundation of our derivative-based regularization strategy. Therefore, the rendering loss can be written as:

$$\mathcal{L}_{img/mask} = \frac{1}{n}\sum_t^n \mathcal{L}_1(\mathcal{R}_{img/mask}(\mathcal{P}_t, \mathbf{F_t}), I_t/M_t), \qquad (11)$$

where $\mathcal{R}_*$ refers to rendering functions, $\mathcal{P}_t, \mathbf{F}_t$ are simulation particle states, $I_t, M_t$ are image and mask groundtruth.

The derivatives of the deformation field to time and spatial coordinates are essentially the particle velocities $\mathbf{v} = \frac{\partial \mathbf{X}}{\partial t}$ and particle deformation gradients $\mathbf{F} = \frac{\partial \mathbf{X}}{\partial \mathbf{x}}$. As for the former, we utilize Gaussian Flow (Gao et al., 2024) to obtain the flow map for each camera view, and use it to supervise the flow obtained by simulation, as in Eq. 12. In this context, the Gaussian flow acts as an implicit supervision for particle velocity. This not only results in a sharper silhouette, serving as an implicit shape constraint, but more importantly, allows for precise supervision within the silhouette.

$$\mathcal{L}_{flow} = \frac{1}{n}\sum_t^n \mathcal{L}_1\left(\mathcal{R}_{gf}(\mathcal{P}_t, \mathbf{F_t}), (\mathcal{R}_{gf}(\mathbf{x}_t, \mathbf{A_t}))\right), \quad (12)$$

where $\mathcal{R}_{gf}(\cdot, \cdot)$ represents the Gaussian flow maps rendered from the Gaussian splats updated by the simulation results. $\mathbf{x}_t = \mathbf{x}_0 + \delta\mathbf{x}_t$, $\mathbf{A}_t$ is the covariance matrix derived from $\delta\mathbf{s}_t$ and $\delta\mathbf{q}_t$ from Eq. 2.

Then we introduce how we guide the spatial derivative of deformations. As stated above, the updated $\mathbf{A}_t$ can be obtained from the dynamic reconstruction results. $\mathbf{A}_0$ is the initial covariance and fixed for all timestamps. Intuitively, if we can solve $\mathbf{F}$ from this, we can build point-wise supervision for the deformation gradient. This equation can be interpreted as a similarity transformation from $\mathbf{A}_0$ to $\mathbf{A}_t$. Due to the high degrees of freedom, deriving an explicit expression for $\mathbf{F}$ is challenging. Additionally, since the solution to a similarity transformation is not unique, the scale of the resulting $\mathbf{F}$ may differ. To solve this, we supervise the rotational and scaling components of $\mathbf{F}$ separately. For the scaling component, we focus only on the relative scale, disregarding the absolute scale. To sum up, we decompose $\mathbf{F}$ into $\Lambda$ and $\mathbf{U}, \mathbf{V}$ via SVD, and define the scaling and rotation regularization respectively.

$$L_s = \mathbb{E}_t(norm(\frac{\delta\mathbf{S}_t}{\mathbf{S}^0 + \epsilon}) - norm(\Lambda - I)). \qquad (13)$$

This regularization constrains the relative magnitude of scaling around the vicinity of particle $p$ in the simulation. The rotational components can be regularized as follows:

$$L_r = \mathbb{E}_t(||\delta\mathbf{R}_t - \mathbf{U}_t\mathbf{V}_t^T||_F^2), \qquad (14)$$

where $\delta\mathbf{R}_t$ is derived from $\delta\mathbf{q}$ in Eq. 2. $U_p^t$ and $V_p^{tT}$ are extracted from SVD of $\mathbf{F}$ on particle $p$ at time $t$. $||.||_F$ represents the Frobenius norm. Further discussion can be seen in **Appendix** J.

As stated in Heterogeneous modeling of Sec. 3.1, we encourage the output of the local material implicit field to have zero mean and minimal variance via:

$$\begin{aligned} \mathcal{L}_{\text{het}} = {} & \lambda_\mu \left\| \mathbb{E}[\eta(\mathbf{x})] \right\|_2^2 \\ & + \lambda_{\text{var}} \max\left(0, \, \text{Var}[\eta(\mathbf{x})] - 0.5^2\right). \end{aligned} \qquad (15)$$

This loss enforces an unbiased overall distribution. The global parameter governs the average response, and the local field contributes only subtle refinements. The total loss is: $\mathcal{L}_{total} = \mathcal{L}_{flow} + \beta_1(\mathcal{L}_{img} + \mathcal{L}_{msk}) + \beta_2(\mathcal{L}_r + \mathcal{L}_s) + \beta_3\mathcal{L}_{het}$. The final optimization target is:

$$\Psi, \Theta, \Phi, \mathbf{P}, \eta = \arg\min_{\Psi, \Theta, \Phi, \mathbf{P}, \eta} \mathcal{L}_{\text{total}}. \qquad (16)$$

## 4. Experiments

**Dataset.** We thoroughly evaluate our method on both synthetic and real-world datasets. For synthetic evaluation, we use the dataset released by PAC-NeRF (Li et al., 2023), which provides multi-view RGB sequences together with 3D ground-truth geometry trajectories across diverse materials. This synthetic benchmark allows us to quantitatively evaluate 3D trajectory accuracy, prove the dynamics grounding, and to verify that our residual stress corrections do not introduce spurious anisotropy when an isotropic backbone is sufficient. Current research still relies heavily on synthetic data due to the scarcity of real-world multi-view dynamic datasets. To fill this gap, we collect a real-world multi-view dataset using an advanced light-field capture system with synchronized industrial cameras. The dataset contains 7 objects, including four elastic objects (Mandarin, Chick1, Chick2, Peanut), two elastoplastic objects (Rainbowball, Rabbit), and one plastic object (Gorilla). Objects are released from random heights to fall vertically onto a platform, and 12 cameras capture synchronized surround-view RGB sequences. More details are provided in **Appendix** F.

**Baselines and Metrics**. We compare our method with PAC-NeRF (Li et al., 2023), DEL (Wang et al., 2024a), GIC (Cai et al., 2024), NeuMA (Cao et al., 2025), and Vid2Sim (Chen et al., 2025a). PAC-NeRF, GIC, and Vid2Sim estimate global physical parameters under classic physical models, while DEL and NeuMA learn dynamics operators with neural components. We evaluate on two settings: object dynamics grounding on the PAC-NeRF dataset and initial-state generalization on our real-world dataset. We report Chamfer

*Table 1.* Dynamic Grounding on PAC-NeRF dataset. All metrics are scaled by 100 for clarification.

| Methods | torus CD↓ | torus EMD↓ | cat CD↓ | cat EMD↓ | playdoh CD↓ | playdoh EMD↓ | droplet CD↓ | droplet EMD↓ | Cream CD↓ | Cream EMD↓ | Bird CD↓ | Bird EMD↓ | Letter CD↓ | Letter EMD↓ | Mean CD↓ | Mean EMD↓ |
|---|---|---|---|---|---|---|---|---|---|---|---|---|---|---|---|---|
| PAC | 21.8 | 11.6 | 9.8 | 14.4 | 18.6 | 5.6 | 10.4 | 3.2 | 20.5 | 12.7 | 19.3 | 21.1 | 12.8 | 8.5 | 16.2 | 11.0 |
| DEL | 21.7 | 10.7 | 7.9 | 12.8 | 12.2 | 2.5 | 9.8 | 1.7 | 19.8 | 9.8 | 17.8 | 20.2 | 12.6 | 7.2 | 14.5 | 9.3 |
| GIC | 20.2 | 9.9 | 7.6 | 12.6 | 12.3 | 2.5 | 10.2 | 1.9 | 19.5 | 10.1 | 16.5 | 19.5 | 10.3 | 7.5 | 13.8 | 9.1 |
| Ours | **20.1** | **9.8** | **7.3** | **12.4** | **11.4** | **2.3** | **9.6** | **1.5** | **19.4** | **9.5** | **16.3** | **19.2** | **10.1** | **6.6** | **13.5** | **8.8** |

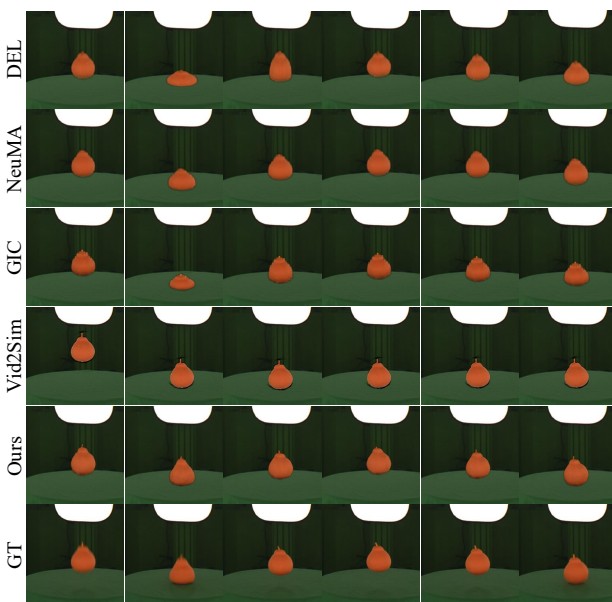

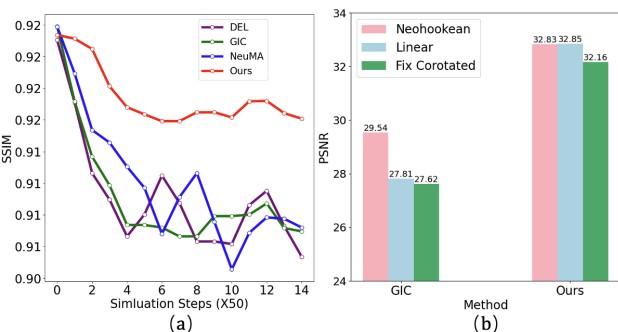

*Figure 4.* Comparisons of long-term dynamics (a) and sensitivity of physical laws (b).

*Figure 3.* Qualitative Comparisons between our methods and baselines. More Visualization can be seen in the **Appendix**.

Distance (CD) and Earth Mover Distance (EMD) for 3D geometry error, and PSNR and SSIM for rendering quality. For PSNR and SSIM, we crop images around the object to avoid background dominance. On the real-world dataset, we report only PSNR and SSIM since ground-truth 3D particle trajectories are unavailable.

**Implementation Details**. We use the same loss weights for all experiments: $\beta_1 = 1.0$, $\beta_2 = 0.1$, $\beta_3 = 0.1$, $\lambda_\mu = 1.0$, and $\lambda_{var} = 1.0$. Here, the image and flow losses provide the primary supervision, while the deformation regularizers and the heterogeneity constraint serve as auxiliary guidance.

### 4.1. Results on Synthetic Data

**Object Dynamics Grounding**. We evaluate all methods on the object dynamics grounding task using the PAC-NeRF dataset. The goal is to assess how accurately each approach captures object dynamics under multi-view video supervision. Following common practice, all methods first reconstruct the initial object shape at the first timestamp using 3DGS, and then simulate the deformation with their corresponding dynamics modules. Quantitative results are reported in Table 1, where our method achieves the best overall performance.

### 4.2. Real-world Generalization

Real-world evaluation is essential because, even when an isotropic backbone is a reasonable approximation, common objects still exhibit mild anisotropy and heterogeneity that can dominate the remaining real-to-sim error. In this experiment, we learn dynamics from video recordings and evaluate generalization on newly captured sequences with different initial orientations and drop heights. We render the simulated results into each camera view and compare them with the held-out footage to measure how well each method transfers beyond its training data.

Quantitative results are shown in Table 2, and qualitative comparisons appear in Fig. 3. Additional visualizations and videos are available in the Appendix and Sup. Mat. Our method consistently outperforms all baselines across material types in real-world settings. Fully neural approaches such as DEL perform worst, likely because modeling full state transitions with large networks overfits under limited real data. NeuMA improves a pretrained NCLaw prior via LoRA, but degrades when the pretraining distribution differs from our real materials; moreover, both NCLaw and NeuMA assume isotropy, limiting their ability to capture anisotropic behaviors. Among system identification methods, GIC performs best, likely due to its use of geometry and motion cues from dynamic 3DGS. Vid2Sim performs poorly, likely because its forward predictor is trained only on synthetic data and does not generalize to our real-world scenes. Overall, our residual stress adaptation remains strong on real data, indicating a high performance ceiling for real-to-sim dynamics learning.

*Table 2.* Quantitative comparisons of the initial state generalizations on the real-world dataset. All SSIM is scaled by 100.

| Methods | Chick1 | | Gorilla | | Mandarin | | Chick2 | | Peanut | | Rabbit | | RBball | | Mean | |
|---|---|---|---|---|---|---|---|---|---|---|---|---|---|---|---|---|
| | PSNR↑ | SSIM↑ | PSNR↑ | SSIM↑ | PSNR↑ | SSIM↑ | PSNR↑ | SSIM↑ | PSNR↑ | SSIM↑ | PSNR↑ | SSIM↑ | PSNR↑ | SSIM↑ | PSNR↑ | SSIM↑ |
| DEL | 28.32 | 91.6 | 29.11 | 90.4 | 31.92 | 92.4 | 28.59 | 90.9 | 28.38 | 91.3 | 28.57 | 91.4 | 30.95 | 91.7 | 29.41 | 91.4 |
| NeuMA | 30.73 | 92.4 | 29.78 | 91.1 | 31.85 | 92.4 | 28.92 | 91.0 | 30.70 | 91.8 | 28.30 | 92.3 | 30.74 | 91.1 | 30.00 | 91.7 |
| GIC | 30.93 | 92.5 | 29.75 | 91.0 | 29.54 | 91.6 | 28.17 | 90.9 | 32.88 | 91.3 | 28.08 | 91.3 | 30.78 | 91.7 | 30.02 | 91.5 |
| Vid2Sim | 26.71 | 90.9 | 29.02 | 90.0 | 25.85 | 89.8 | 25.70 | 89.5 | 30.69 | 94.6 | 26.85 | 90.3 | 31.71 | 91.7 | 28.08 | 91.0 |
| Ours | **32.05** | **92.7** | **30.19** | **91.9** | **32.83** | **92.7** | **30.17** | **91.6** | **33.01** | **92.0** | **30.35** | **92.1** | **32.06** | **92.7** | **31.35** | **92.3** |

*Table 3.* Ablation studies on three real-world scenes

| | Rabbit | | Gorilla | | Rainbowball | |
|---|---|---|---|---|---|---|
| | PSNR↑ | SSIM↑ | PSNR↑ | SSIM↑ | PSNR↑ | SSIM↑ |
| no $\mathbb{C}$ | 28.42 | 91.3 | 29.66 | 91.0 | 30.67 | 91.8 |
| no Het | 29.67 | 91.5 | 29.98 | 91.2 | 31.15 | 91.8 |
| no $\mathbf{L}^{pre}$ | 29.97 | 91.7 | 30.12 | 91.3 | 31.18 | 91.9 |
| no $\mathbf{L}^{post}$ | 29.88 | 91.7 | 30.18 | 91.6 | 31.21 | 91.6 |
| no $\mathcal{L}_{flow}$ | 29.13 | 91.6 | 29.83 | 90.9 | 30.29 | 91.0 |
| no $\mathcal{L}_{scale}$ | 30.05 | 91.9 | 30.12 | 91.1 | 30.84 | 92.1 |
| no $\mathcal{L}_{rot}$ | 30.14 | 91.3 | 29.99 | 91.2 | 31.98 | 92.0 |
| Full mode | **30.35** | **92.1** | **30.19** | **91.9** | **32.06** | **92.7** |

## 4.3. Additional Experiments and Analysis

**Ablation Studies**. We conduct ablations on three real-world examples (rabbit, gorilla, and rainbowball) to assess the contribution of each component. Results are summarized in Table 3. We remove key terms in the stress adapter, including $\mathbb{C}$, $\mathbf{L}^{pre}$, $\mathbf{L}^{post}$, and the heterogeneity modeling (no Het). The redistribution term $\mathbb{C}$ has the largest impact, confirming the importance of anisotropic stress redistribution, while the linear correction terms also provide consistent gains. The heterogeneity module is also important. Removing it leads to consistent performance drops across all three scenes: PSNR decreases from 30.35 to 29.67 on rabbit, from 30.19 to 29.98 on gorilla, and from 32.06 to 31.15 on rainbowball. This shows that controlled spatial variation complements residual anisotropic stress adaptation and improves real-world dynamics modeling.

We further ablate the motion constraints, including the flow loss and the scale and rotation losses. All these terms improve accuracy and generalization, confirming the importance of derivative-level supervision for stable learning from videos.

**Predictions of Long-term Dynamics**. We present comparisons of long-term simulation in Fig. 4 (a), which include the SSIM at each timestep for the "rabbit" scenario. The comparison demonstrates that our method consistently outperforms the others throughout the entire simulation. Additionally, our method exhibits a much slower increase in loss over time compared to the other approaches, maintaining better stability in long-term predictions.

**Robustness to Prior Physical Models**. We study how the choice of prior physical law affects performance on Chick1. As shown in Fig. 4(b), parameter-calibration baselines are sensitive to the selected model, since they can only adjust

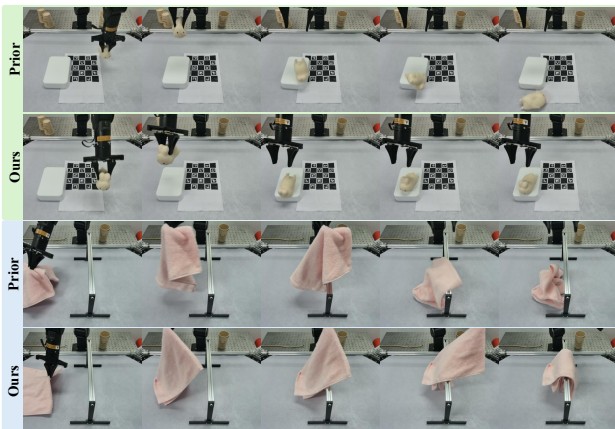

*Figure 5.* Applications of zero-shot robot manipulation transfer.

parameters to compensate for model errors. In contrast, our method refines the constitutive prior itself, making it less dependent on model selection. GIC exhibits large performance variation across different priors, whereas our method remains stable. Notably, even starting from a simple linear stress–strain model, our approach can progressively correct the prior and achieve higher PSNR, demonstrating strong robustness and flexibility.

**Application on Downstream Manipulation Tasks**. We further evaluate whether learning more accurate real-world dynamics benefits downstream robot manipulation. Specifically, we define two manipulation tasks and first learn their object dynamics from video interactions. We then deploy the learned dynamics in simulation to train a policy, and directly transfer the policy to the real robot for evaluation. As a baseline, we use a purely isotropic prior physical model in the simulator. As shown in Fig. 5, our learned-dynamics policies achieve substantially higher real-world success rates: on the elastic-rabbit placement task (placing a deformable rabbit onto a white box), our policy succeeds in **68 trials** per 100, compared to **42**% for the isotropic prior; on the tower-hanging task, our policy succeeds in **82 trials**, compared to **55**% for the isotropic baseline. These results suggest that accounting for continuum residual effects can be a key factor for further improving real-to-sim dynamics and sim-to-real transfer.

**Sensitivity to Dynamic Reconstruction Quality.** Our motion constraints rely on dynamic reconstruction cues, so we further evaluate their sensitivity to reconstruction quality. Specifically, we train MoSA using dynamic 3DGS recon-

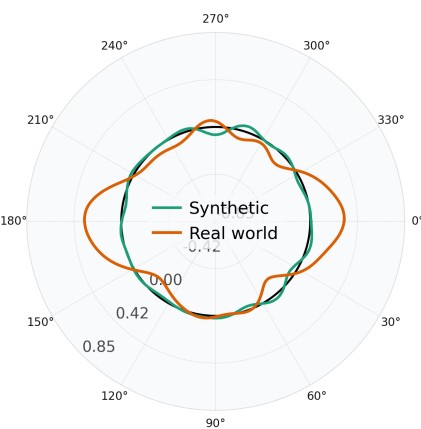

*Figure 6.* Analysis of learned anisotropy with directional Jacobian

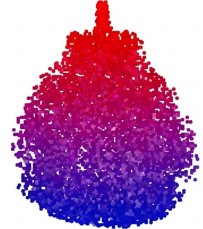

*Figure 7.* Spatial visualization of the normalized learned heterogeneity field $\eta(\mathbf{x})$ on test objects. Reddish regions correspond to larger learned local stiffness, while bluish regions indicate softer regions.

*Table 4.* Sensitivity to dynamic reconstruction quality on the Rabbit scene. R.Steps denotes the progress of reconstruction.

| R.Steps | PSNR↑ | SSIM↑ | ΔPSNR |
|---------|-------|-------|-------|
| 5k      | 29.30 | 91.6  | +0.17 |
| 7k      | 29.65 | 91.7  | +0.52 |
| 10k     | 29.95 | 91.8  | +0.82 |
| 15k     | 30.15 | 91.9  | +1.02 |
| 20k     | 30.28 | 92.0  | +1.15 |
| 30k     | 30.35 | 92.1  | +1.22 |

structions obtained at different optimization stages on the Rabbit scene, and report the downstream simulation performance rather than the reconstruction quality itself. As shown in Table 4, motion constraints consistently improve performance over the model without motion constraints, whose PSNR is 29.13. Even early-stage reconstructions provide positive gains, indicating that the proposed motion constraints benefit downstream physics learning instead of simply amplifying reconstruction errors.

**Further Analysis of What the Model Learns**. We analyze whether our residual stress adaption module learns physically meaningful effects rather than overfitting a single sequence. Since PAC-NeRF is generated by an isotropic simulator, it contains no true anisotropy; thus our model should not introduce spurious directionality. We define the directional Jacobian response of Eq. 3: $w(\theta) = \left\| \frac{\partial \hat{\boldsymbol{\sigma}}}{\partial \boldsymbol{\sigma}^{iso}} \cdot \text{vec}(\mathbf{n} \otimes \mathbf{n}) \right\|_2$, which characterizes the stress redistribution trend induced by the Progressive Stress Adaptation module. Fig. 6 shows that on PAC-NeRF, $w(\theta)$ stays near zero with no clear directional pattern, indicating only mild adjustments to the isotropic prior. In contrast, on real data (Mandarin), $w(\theta)$ exhibits pronounced directionality aligned with the object axis, suggesting that stress adaptation module adaptively captures missing anisotropic effects when present. More analysis is provided in Appendix A.

**Spatial Variation of the Learned Heterogeneity**. We

visualize the normalized learned $\eta(\mathbf{x})$ on three held-out test objects in Fig. 7. Since $\eta(\mathbf{x})$ modulates the global material parameter locally, its spatial distribution reflects learned material heterogeneity. The resulting fields show smooth and object-dependent patterns, with reddish regions corresponding to higher local stiffness. This confirms that our continuous field learns physically meaningful spatial variations, rather than merely fitting unstructured residual noise.

More experimental analysis including system identification, future prediction, etc., are reported in our Appendix.

## 5. Conclusion

In conclusion, our results indicate that further reducing the real-to-sim gap for everyday deformable objects is less about re-learning the dominant isotropic backbone model, and more about capturing the mild but systematic continuum residual effects caused by the anisotropy and heterogeneity that standard simulators ignore. By explicitly constraining learning to residual stress corrections and strengthening supervision with motion cues, our approach achieves consistently better stability and transfer in both dynamics metrics and downstream robot manipulation. We hope this work encourages future dynamics learning methods to combine strong physical backbones with targeted residual modeling for scalable real-world deployment.

## Impact Statement

MoSA improves the fidelity of real-to-sim continuum dynamics by modeling mild residual anisotropy and heterogeneity beyond standard isotropic assumptions. This can benefit a broad range of applications that rely on reliable physical simulation, including graphics, embodied interaction, and robotics, by enabling more accurate and data-efficient dynamics identification and reducing failures caused by model mismatch. In the long term, stronger simulation reliability may support safer design, testing, and deployment of systems that interact with the physical world,

and may facilitate higher-fidelity physics-consistent digital twins.

As with many advances in physical modeling, improved simulation can be used in both beneficial and potentially harmful ways, including in autonomous systems. Our work focuses on foundational algorithmic improvements for dynamics learning and does not involve sensitive personal data or human subjects. We encourage responsible use and evaluation in safety-critical settings.

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

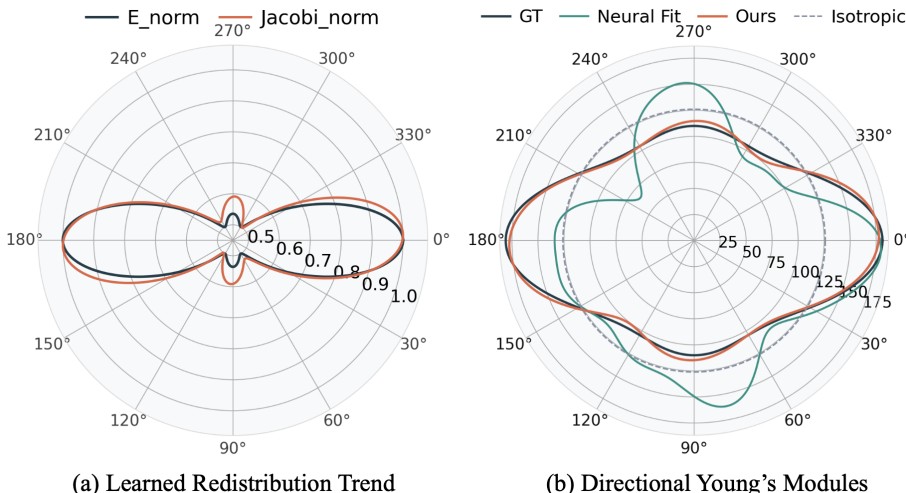

(a) Learned Redistribution Trend      (b) Directional Young's Modules

*Figure 8.* Physically grounded analysis of learned anisotropy

## A. Verification of Learned Anisotropy Knowledge

To verify that the proposed physics-informed **Progressive Stress Adaptation** module truly learns the underlying physical anisotropy rather than merely benefiting from increased parameters, we design two validation experiments based on the same simulation. A uniaxial compression test is conducted on a cylindrical specimen with a prescribed strong XY-plane anisotropy in its elastic modulus. During loading, deformation videos are recorded to evaluate the learned mechanical behavior.

First, we analyze the Jacobian matrix in Eq. 3. Physically, this Jacobian characterizes the *internal stress redistribution trend* learned by the stress adaptation module. By scanning directions on the XY plane, we compute the directional Jacobian response $w(\theta) = \left\| \frac{\partial \hat{\boldsymbol{\sigma}}}{\partial \boldsymbol{\sigma}^{iso}} \cdot \text{vec}(\mathbf{n} \otimes \mathbf{n}) \right\|_2$, and compare its pattern with the GT anisotropic modulus $E_{GT}(\theta)$, as shown in Fig. 8(a). The two curves exhibit that the learned redistribution trend closely aligns with the physical stiffness anisotropy. Next, we derive the effective directional modulus from the model-predicted stress–strain relations and compare it with two baselines: (i) an isotropic model and (ii) a neural model that directly learns the anisotropic mapping, i.e., $\hat{\boldsymbol{\sigma}} = NN(\boldsymbol{\sigma}^{iso}, \mathbf{U}, \theta)$. As shown in Fig. 8 (b), our model (orange line) accurately reproduces the GT anisotropic modulus (black line), while the baselines (grey and green lines) fail to capture the directional dependence. These results confirm that the superior performance of the stress adapter primarily arises from its physics-aware design rather than from a mere increase in network parameters.

## B. Evaluation on Spring-GS dataset

To further indicate the superiority of our method, we follow the setting in Spring-GS to use 20 frames as training data to predict the rest 10 frames. We first train these models the training data, then use the trained models or estimated parameters to simulate the full trajectory. We report the results in Table 5. It is observed that our method stably performs better than other counterparts. The performance of NeuMA may not be as expected because the data used for its pretraining has a significant gap compared to the Spring-GS dataset.

*Table 5.* Quantitative Comparisons of the Future State Prediction on Spring-GS dataset.

| | CD↓ | | | | EMD↓ | | | | PSNR↑ | | | | SSIM↑ | | | |
|---|---|---|---|---|---|---|---|---|---|---|---|---|---|---|---|---|
| Method | SpGS | NeuMA | GIC | Ours | SpGS | NeuMA | GIC | Ours | SpGS | NeuMA | GIC | Ours | SpGS | NeuMA | GIC | Ours |
| torus | 2.38 | 1.68 | 0.75 | **0.68** | 0.087 | 0.043 | 0.034 | **0.032** | 16.83 | 17.83 | 20.24 | **20.75** | 0.919 | 0.913 | 0.942 | **0.949** |
| cross | 1.57 | 2.12 | 1.09 | **1.02** | 0.051 | 0.059 | 0.058 | **0.049** | 16.93 | 23.52 | 30.51 | **30.76** | 0.940 | 0.928 | 0.939 | **0.941** |
| cream | 2.22 | 1.01 | 0.94 | **0.93** | 0.094 | 0.050 | 0.050 | **0.046** | 15.42 | 18.96 | 19.15 | **19.22** | 0.862 | 0.877 | 0.909 | **0.912** |
| apple | 1.87 | 0.16 | 0.22 | **0.15** | 0.076 | 0.029 | 0.030 | **0.028** | 21.55 | 26.35 | 26.89 | **27.16** | 0.902 | 0.927 | 0.948 | **0.950** |
| paste | 7.03 | 6.22 | 2.79 | **2.61** | 0.126 | 0.097 | 0.096 | **0.094** | 14.71 | 16.81 | 16.31 | **16.95** | 0.872 | 0.896 | 0.894 | **0.903** |
| chess | 2.59 | 2.12 | 0.77 | **0.83** | 0.095 | 0.113 | 0.059 | **0.056** | 16.08 | 16.37 | 18.44 | **19.15** | 0.881 | 0.838 | 0.912 | **0.916** |
| banana | 18.50 | 0.58 | 0.12 | **0.11** | 0.135 | 0.093 | 0.017 | **0.015** | 17.89 | 22.08 | 29.29 | **29.39** | 0.904 | 0.913 | 0.964 | **0.968** |

## C. Network Architecture

As we stated in the main text, we adopt four lightweight neural networks to separately model the three neural corrected modules. $\Phi$ and $\Theta$ networks aim to produce the two linear adjustment term. $\Psi_1$ and $\Psi_2$ networks generate the four redistribution matrix which are decomposed from the fourth-order tensor contraction $\mathbb{C}$. All subnetworks include two hidden layers, each with 64 hidden dimensions. The $U$ in each input is the Right Polar matrix of the deformation gradient $\mathbf{F}$, which is a symmetric matrix. Hence we only input the upper triangle of the $U$ (6 independent components) to these networks. We adopt the singular value decomposition to compute the $U$. Similarly, we include the prior stress into the input tensor, which is also symmetric.

The output of each network is a 12-dimensional vector, where the first six elements and the last six elements form an upper triangular matrix and a lower triangular matrix, respectively. These two matrices are then each added to an identity matrix and multiplied together to obtain the final output. According to LU decomposition, this ensures that the output remains an invertible matrix with desirable properties.

## D. Parameter Search for the $\alpha$

In this section, we discuss the selection of hyperparameters $\alpha_1$ and $\alpha_2$ from the Equation 7 in our main paper. To guide our selection, we performed a coarse parameter search with a step size of 0.1 in the Chick1 scenario, as shown in the Fig. 9. The heatmap illustrates the variation in PSNR values based on different combinations of $\alpha_1$ and $\alpha_2$, where darker colors correspond to higher PSNR values.

From this exploration, we observed that the optimal settings for both $\alpha_1$ and $\alpha_2$ lie around 0.15, which is where we chose to set them. Even with these relatively coarse settings, the PSNR remains high. Notably, even when the values of both $\alpha_1$ and $\alpha_2$ are set to 0.5, the PSNR still reaches 31.29, which is higher than the baseline method. This suggests that our method is not overly sensitive to small changes in the values of $\alpha_1$ and $\alpha_2$, and reasonably chosen settings still yield competitive results.

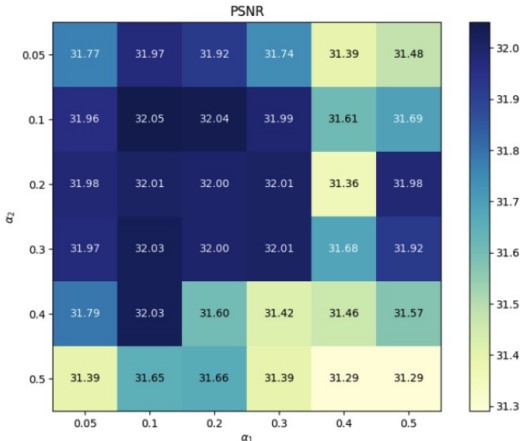

*Figure 9.* Grid Search for the hyperparameter $\alpha_{1,2}$

## E. Effect of the Prior Stress

We evaluate the effect of removing prior stress, specifically $\sigma(\epsilon)_{kl}$ in Eq. 3 of the main paper, and relying only on $\mathbb{C}$ and $L$ (as shown in Table 3). While the "no prior" model fits the training data well, its test performance decreases, indicating overfitting. In contrast, incorporating prior stress improves generalization by facilitating optimization, as evidenced by the higher PSNR and SSIM values in both training and testing for our method.

Notably, even when prior stress is omitted, our method still outperforms isotropic models like GIC and DEL (shown in Table 3 of the main paper). This highlights the importance of anisotropic modeling in capturing more accurate and generalizable dynamics. The results in Table 6 demonstrate that, while prior stress enhances model performance, our method's ability to capture complex dynamics without it still surpasses simpler isotropic models.

*Table 6.* Comparisons between with and without prior stress

| Episode | Method | Chick1 | | Chick2 | |
|---|---|---|---|---|---|
| | | PSNR↑ | SSIM↑ | PSNR↑ | SSIM↑ |
| Train | NoPriorStress | 33.15 | 0.937 | 30.82 | 0.921 |
| | Ours | **33.81** | **0.938** | **30.85** | **0.922** |
| Test | NoPriorStress | 31.03 | 0.924 | 28.99 | 0.911 |
| | Ours | **32.05** | **0.927** | **30.17** | **0.916** |

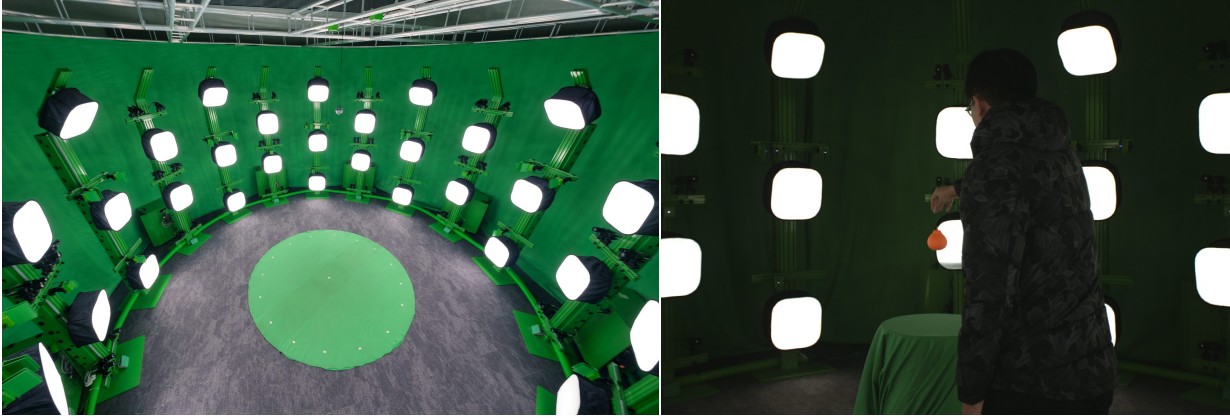

(a) Light field reconstruction system      (b) Release objects from predetermined height

*Figure 10.* Visual illustration of our real-world data collection pipeline

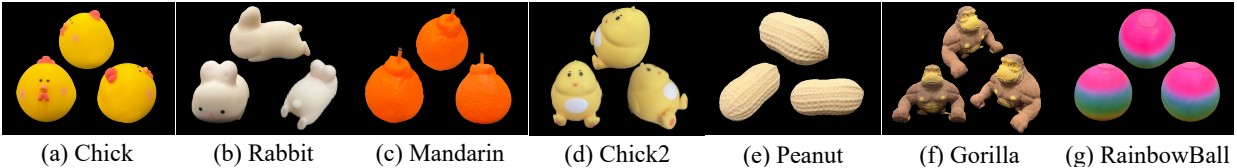

(a) Chick    (b) Rabbit    (c) Mandarin    (d) Chick2    (e) Peanut    (f) Gorilla    (g) RainbowBall

*Figure 11.* The visualizations of all objects in our real-world dataset

## F. Real-World Dataset

### F.1. Data Collection

To achieve high-quality, real-world dynamic data, we utilize an advanced light field reconstruction system, as shown in Fig.10, to gather multi-view RGB sequences. This system is equipped with ultra-high-precision industrial cameras arranged in a spherical configuration, allowing for synchronized, surround-view data collection with a delay of less than 1 ms. During the data collection phase, we release the objects to fall vertically from a predetermined height onto a platform, with the light field reconstruction system capturing synchronized multi-view data. After capturing the footage, we conduct a meticulous frame-by-frame inspection to ensure there are no flips or occlusions during the fall. We employ 12 to 15 industrial cameras, each capturing a sequence of 15 RGB frames from different fixed viewpoints. This approach ensures the acquisition of high-quality, realistic, and complete free-fall dynamics.

After collecting the initial data, we proceeded with post-processing the raw images. The images are cropped to remove redundant static elements from the background. This ensured that the focus remained solely on the dynamic aspects of the captured data. After that, we employed GroundingSAM techniques to generate precise masks for each frame, facilitating the accurate separation of the foreground and background.

### F.2. Objects Description

Given that the dataset is specifically designed to capture real-world dynamics during free fall, we carefully select seven distinct objects, each with unique physical properties, to represent a wide range of materials, as illustrated in Fig. 11. These objects are:

**Gorilla**: This object contains deformable materials such as space sand, which undergo irreversible plastic deformations during free fall. Unlike elastic objects will return to their original shape after deformation, Gorilla exhibits permanent, non-recoverable changes. We select this object to demonstrate the superior performance of our model in learning the physical dynamics of plastic material only from visual data.

**Chick1, Chick2, Peanut, and Mandarin**: These objects are elastic but differ in their levels of softness and elasticity and vary in their textures and shapes. By including a variety of elastic materials with distinct properties, we ensure that our dataset captures a broad spectrum of deformation behaviors, which is essential for understanding diverse physical dynamics.

**Rainbowball, Rabbit**: These objects are made of elastoplastic materials, and exhibit both elastic and plastic behaviors. When subjected to forces during free fall, these objects initially deform elastically, returning to their original shape when the stress is small. However, they undergo plastic deformations that are irreversible when the force is beyond a certain threshold, similar to purely plastic materials. Rainbowball and Rabbit represent a more complex material behavior that bridges the gap between purely elastic and purely plastic materials.

## G. Effect and Ablation of the Prior Stress

We evaluate the effect of removing prior stress, specifically $\boldsymbol{\sigma}(\epsilon)_{kl}$ in Eq. 3 of the main paper, and relying only on $\mathbb{C}$ and $L$ (as shown in Table 3). While the "no prior" model fits the training data well, its test performance decreases, indicating overfitting. In contrast, incorporating prior stress improves generalization by facilitating optimization, as evidenced by the higher PSNR and SSIM values in both training and testing for our method.

Notably, even when prior stress is omitted, our method still outperforms isotropic models like GIC and DEL (shown in Table 3 of the main paper). This highlights the importance of anisotropic modeling in capturing more accurate and generalizable dynamics. The results in the following Table demonstrate that, while prior stress enhances model performance, our method's ability to capture complex dynamics without it still surpasses simpler isotropic models.

*Table 7.* Comparisons between with and without prior stress

| Episode | Method | Chick1 | | Chick2 | |
| --- | --- | --- | --- | --- | --- |
| | | PSNR↑ | SSIM↑ | PSNR↑ | SSIM↑ |
| Train | NoPriorStress | 33.15 | 0.937 | 30.82 | 0.921 |
| | Ours | **33.81** | **0.938** | **30.85** | **0.922** |
| Test | NoPriorStress | 31.03 | 0.924 | 28.99 | 0.911 |
| | Ours | **32.05** | **0.927** | **30.17** | **0.916** |

## H. Complexity Analysis with Other Learned Simulators

We conducted a complex analysis comparing the training and inference times of our method with two other learning-based methods, NeuMA and DEL. As shown in Table 8, our method outperforms the others in terms of both training and inference efficiency. Specifically, our method reduces training time to 3.5 hours, significantly faster than NeuMA (6.5 hours) and DEL (9.0 hours). In terms of inference, our method also leads with only 0.76 seconds per inference, compared to 1.05 seconds for NeuMA and 1.09 seconds for DEL. These results demonstrate that our approach not only achieves superior performance but also operates with greater efficiency, making it a more scalable solution for real-time applications.

*Table 8.* Comparisons of Training and Inference Times.

| | NeuMA | DEL | NoMotion | NoStress | NoFlow | Ours |
| --- | --- | --- | --- | --- | --- | --- |
| Train (h) | 6.5h | 9.0h | 4.5h | 5.8h | 4.3h | **3.5h** |
| Infer (s) | 1.05s | 1.09s | – | – | – | **0.76s** |

## I. Results of System Identification

Although our method is designed to both estimate global physical parameters and refine local constitutive behavior through stress correction, it still achieves superior performance even when evaluating only the accuracy of global parameter estimation. This demonstrates that our approach is not just capable of correcting local dynamics but also highly effective at identifying global material properties. Out of 23 parameter estimations across different scenes, **our method achieved the**

**highest accuracy in 21 cases**.

This improvement can be attributed to two key components in our design. First, the stress correction module simplifies the learning objective by offloading local modeling errors from the global parameter estimation. By refining the prior constitutive model at the local level, it allows the global parameters—such as Young's modulus and Poisson's ratio—to focus on capturing the overall material behavior without being forced to compensate for local discrepancies. Second, our motion-constrained optimization strategy plays a crucial role. By incorporating motion cues from dynamic 3DGS, this strategy not only implicitly supervises particle positions (first-order dynamics), but also regularizes the spatial and temporal derivatives of the deformation field. These richer constraints help guide the optimization toward more physically plausible solutions, leading to more accurate parameter estimation. Overall, the combination of these two components allows our method to outperform baselines, even in standard system identification settings.

*Table 9.* System Identification Performance on PAC-NeRF Dataset. All the values and quantities in the table are scaled based on the magnitude of the ground truth values.

|  | PAC-NeRF | GIC | Ours | Ground Truth |
|---|---|---|---|---|
| Droplet | $\mu, \kappa = 2.09, 1.085$ | $\mu, \kappa = \mathbf{2.01}, 0.18$ | $\mu, \kappa = \mathbf{2.01}, \mathbf{1.07}$ | $\mu, \kappa = 2, 1$ |
| Letter | $\mu, \kappa = 83.85, \mathbf{1.35}$ | $\mu, \kappa = 95.05, 1.00$ | $\mu, \kappa = \mathbf{97.00}, 1.00$ | $\mu, \kappa = 100, 10^5$ |
| Cream | $\mu, \kappa = 1.21, 1.57$ | $\mu, \kappa = 1.03, 1.48$ | $\mu, \kappa = \mathbf{1.01}, \mathbf{1.39}$ | $\mu, \kappa = 1, 1$ |
|  | $\tau_Y, \eta = 3.16, 5.6$ | $\tau_Y, \eta = \mathbf{2.98}, 6.6$ | $\tau_Y, \eta = 2.97, \mathbf{7.75}$ | $\tau_Y, \eta = 3, 10$ |
| Toothpaste | $\mu, \kappa = 6.51, 2.22$ | $\mu, \kappa = 4.19, 9.24$ | $\mu, \kappa = \mathbf{4.52}, \mathbf{9.37}$ | $\mu, \kappa = 5, 10$ |
|  | $\tau_Y, \eta = 228, \mathbf{9.77}$ | $\tau_Y, \eta = 226, 9.1$ | $\tau_Y, \eta = \mathbf{212}, 9.63$ | $\tau_Y, \eta = 200, 10$ |
| Torus | $E, \nu = 1.04, 0.322$ | $E, \nu = \mathbf{0.99}, 0.295$ | $E, \nu = \mathbf{0.99}, \mathbf{0.298}$ | $E, \nu = 1, 0.3$ |
| Bird | $E, \nu = 2.78, 0.273$ | $E, \nu = 3.08, 0.284$ | $E, \nu = \mathbf{3.02}, \mathbf{0.29}$ | $E, \nu = 3, 0.3$ |
| Playdoh | $E, \nu, \tau_Y = 3.84, 0.272, 1.69$ | $E, \nu, \tau_Y = 1.58, 0.322, 1.56$ | $E, \nu, \tau_Y = \mathbf{1.72}, \mathbf{0.289}, \mathbf{1.55}$ | $E, \nu, \tau_Y = 2, 0.3, 1.54$ |
| Cat | $E, \nu, \tau_Y = 1.61, 0.293, 3.57$ | $E, \nu, \tau_Y = \mathbf{0.98}, 0.296, 3.76$ | $E, \nu, \tau_Y = \mathbf{0.98}, \mathbf{0.297}, \mathbf{3.77}$ | $E, \nu, \tau_Y = 1, 0.3, 3.85$ |
| Trophy | $\theta^0_{fric} = 36.1°$ | $\theta^0_{fric} = 38.0°$ | $\theta^0_{fric} = \mathbf{38.5°}$ | $\theta^0_{fric} = 40°$ |

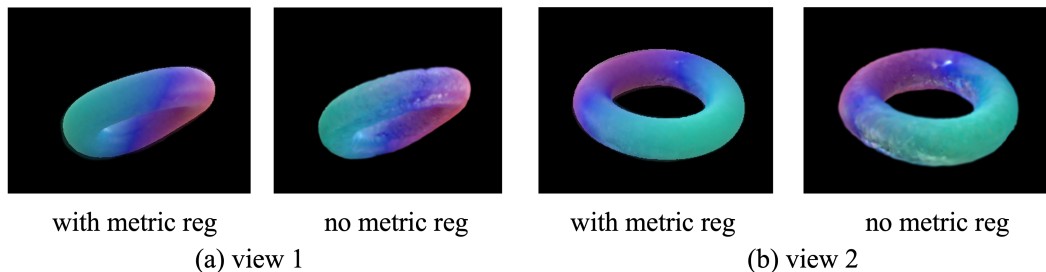

| with metric reg | no metric reg | with metric reg | no metric reg |
|---|---|---|---|
| (a) view 1 | | (b) view 2 | |

*Figure 12.* Comparisons of simulation results on PAC-NeRF dataset with or without metric regularization (e.g. $\mathcal{L}_{scale}, \mathcal{L}_{rot}$)

## J. Explanation of Scale Regularization

In this section, we establish the relationship between the scale changes of Gaussian splats and the simulated deformation gradient. Since the static Gaussians are assumed to be isotropic, all anisotropic deformations during dynamic reconstruction are captured by the deformation network $\mathcal{F}_\theta$ in dynamic 3DGS paradigm introduced in the main paper, Eq. 2. Therefore, our goal is to derive a way to regularize the deformation gradient using the output of this network. Let $\Lambda$ denote the singular values of the deformation gradient $\mathbf{F}$, and let $\delta S$ be the scale change predicted by the deformation network. Based on this, we derive the following:

$$\Lambda S_0 := S_0 + \delta S$$
$$(\Lambda - I) := \frac{\delta S}{S_0} \tag{17}$$

*Table 10.* Quantitative comparison on the public real-world subset of Spring-GS. SSIM values are scaled by 100.

| Method | PSNR↑ | | | | SSIM↑ | | | |
|---|---|---|---|---|---|---|---|---|
| | Bun | Burger | Dog | Pig | Bun | Burger | Dog | Pig |
| Spring-GS | 30.69 | 34.01 | 32.10 | 34.97 | 99.2 | 99.4 | 99.4 | 99.6 |
| NeuMA | 31.27 | 23.78 | 25.61 | 25.40 | 99.4 | 99.3 | 99.5 | 99.5 |
| GIC | 34.68 | 40.45 | 37.17 | 38.32 | 99.5 | 99.7 | 99.7 | 99.7 |
| Ours | **41.12** | **43.25** | **39.21** | **42.16** | **99.7** | **99.8** | **99.8** | **99.9** |

Since we only supervise the relative deformation magnitudes along three directions, we incorporate a normalization function to ensure the scale of the deformation is consistent across these directions.

$$norm(\Lambda - I) := norm(\frac{\delta S}{S_0 + \epsilon}) \tag{18}$$

## K. Qualitative Ablations of the two Deformation Regularization Losses

We present the results of the ablation about the effect of the two deformation regularizations, e.g., **the scale and rotation loss**, in Fig. 12, comparing two cases: one with the two losses and the other without, observed from two different viewpoints. We can observe that the ring-like shape with metric regularization looks more stable, with a uniform and smooth surface. In contrast, the shape without metric regularization shows significant deformation. The inclusion of metric regularization results in a smoother and more consistent geometric shape, reducing distortion and irregularity. This leads to better structural stability and consistency, which is particularly beneficial for modeling or optimization tasks. Therefore, metric regularization proves to be effective in enhancing model performance, especially when dealing with complex geometric forms.

## L. Limitation

Our current pipeline relies on accurate multi-view capture, calibration, and dynamic reconstruction; failures in these steps can weaken the motion cues and reduce training stability. In addition, MoSA is designed to model mild residual anisotropy and heterogeneity beyond an isotropic backbone; cases with extremely strong anisotropy or rapidly varying material properties may require richer priors or additional sensing signals. As future work, we plan to improve robustness to imperfect observations, extend the framework to broader material regimes, and explore incorporating complementary sensors or learned uncertainty to better handle challenging real-world captures.

## M. Additional Qualitative Comparisons

We show additional qualitative comparisons for the other scenarios in the real-world dataset in Fig. 13, Fig. 14, Fig. 15, Fig. 16, Fig. 17, Fig. 18. The images illustrate the effectiveness of our method, as it consistently produces clear, accurate dynamics over time for different objects. Our approach captures the object behavior in a detailed and realistic manner. The fine-grained motion details highlight the robustness and high performance of the proposed method in dynamic learning from real-world data. Vid2Sim performs worse than reported in its original paper, mainly due to several limitations. It is trained on synthetic data and heavily relies on the accuracy of the initial LBS weight predictions. Additionally, it requires strict input formats, such as exactly 16 video frames. In contrast, GIC, as a general-purpose optimization-based method, outperforms Vid2Sim on our real-world dataset. DEL and NeuMA, trained with only a single sequence, show limited generalization during testing. DEL even occasionally fails to maintain basic geometric structure.

## N. Evaluation on the Spring-GS Real-World Subset

To further validate the real-world generalization ability of MoSA, we conduct additional experiments on the public real-world subset of Spring-GS. As shown in Table 10, our method achieves the best performance on all scenes in terms of both PSNR and SSIM, demonstrating that the advantage of MoSA also transfers to public real-world benchmarks.

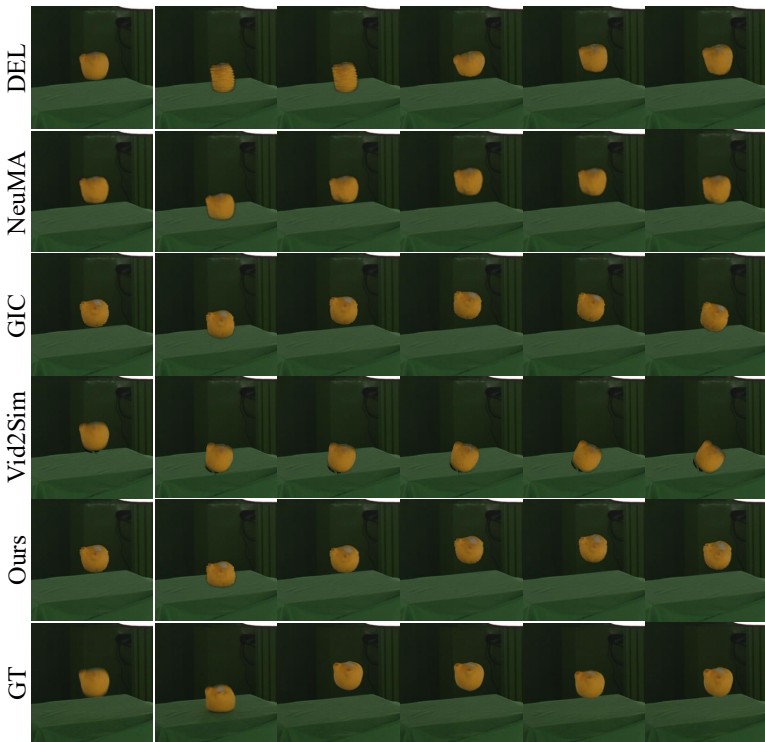

*Figure 13.* Qualitative comparison of different baselines and our methods on the chick2 scenario.

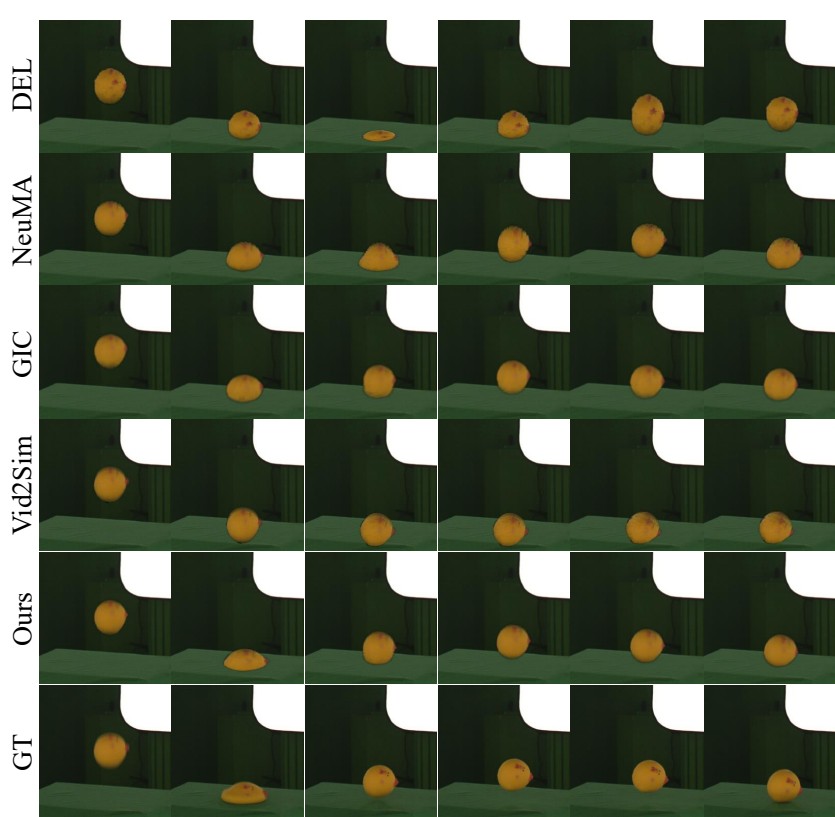

*Figure 14.* Qualitative comparison of different baselines and our methods on the chick1 scenario.

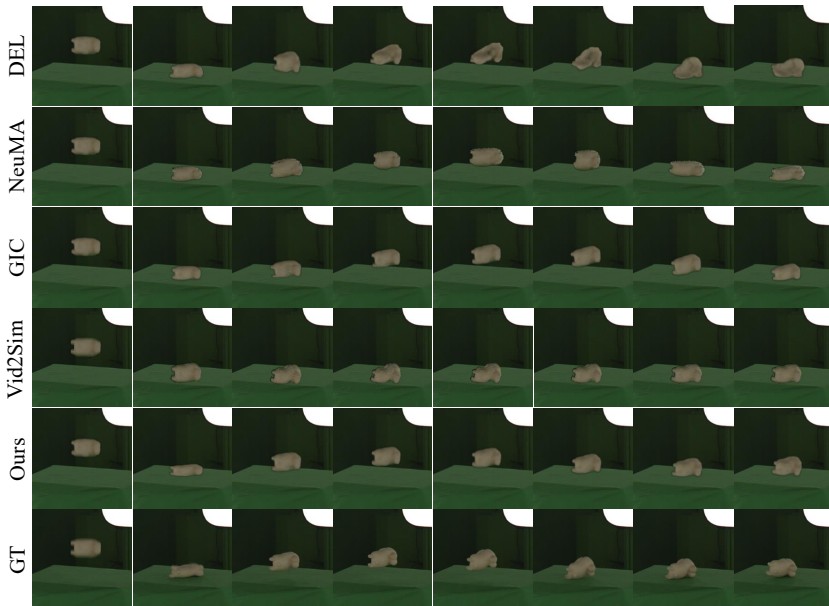

*Figure 15.* Qualitative comparison of different baselines and our methods on the rabbit scenario.

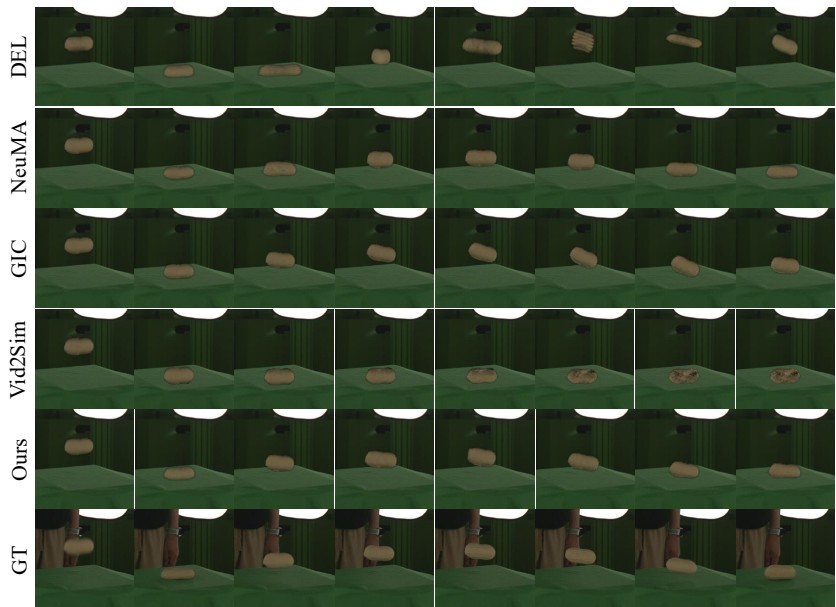

*Figure 16.* Qualitative comparison of different baselines and our methods on the peanut scenario.

## O. Derivation of Microplane Parametrization of the Fourth-Order Correction Operator

In this appendix, we show, in a self-contained way, that the proposed microplane parametrization using $(C_x, C_y, C_z, C_{xyz})$ is mathematically equivalent to using a single fourth-order tensor $C_{ijkl}$ acting on the stress tensor in Eq. (3). The key idea is that all three stages of our Constructions are linear maps on the stress components, and any composition of linear maps is itself a linear map. In Voigt notatio,n this means that the The whole pipeline can always be written as a single $6 \times 6$ matrix, which in turn is equivalent to a fourth-order tensor.

We show that our microplane parametrization with $(C_x, C_y, C_z, C_{xyz})$ is equivalent to a single fourth-order correction tensor as in Eq. 3.

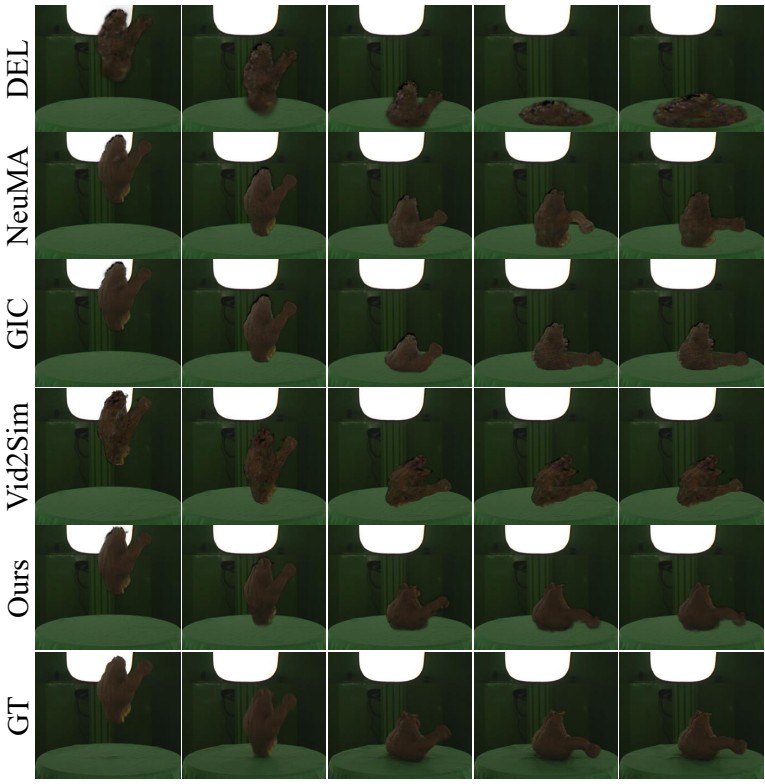

*Figure 17.* Qualitative comparison of different baselines and our methods on the gorilla scenario.

**Fourth-order tensor contraction in Voigt form.** The core correction in Eq. 3 can be written as

$$\hat{\sigma}_{ij} = \sigma_{ij} + \mathbb{C}_{ijkl}\, \sigma_{kl}, \tag{19}$$

where $C_{ijkl}$ is a fourth-order tensor. Because $\sigma_{ij}$ is symmetric, we can collect the six independent components into a Voigt vector

$$\boldsymbol{\sigma} = \begin{bmatrix} \sigma_{xx} \\ \sigma_{yy} \\ \sigma_{zz} \\ \sigma_{yz} \\ \sigma_{zx} \\ \sigma_{xy} \end{bmatrix}, \qquad \hat{\boldsymbol{\sigma}} = \begin{bmatrix} \hat{\sigma}_{xx} \\ \hat{\sigma}_{yy} \\ \hat{\sigma}_{zz} \\ \hat{\sigma}_{yz} \\ \hat{\sigma}_{zx} \\ \hat{\sigma}_{xy} \end{bmatrix}. \tag{20}$$

Then $C_{ijkl}$ is in one-to-one correspondence with a $6 \times 6$ matrix $\mathbf{C}$, and Eq. 19 is exactly the matrix–vector multiplication

$$\hat{\boldsymbol{\sigma}} = (\mathbf{I}_6 + \mathbf{C})\, \boldsymbol{\sigma}, \tag{21}$$

where each row of $(\mathbf{I}_6 + \mathbf{C})$ lists the coefficients of the linear combination that produces one corrected stress component from all six original components. In other words, Eq. 21 is just a linear map $\mathbb{R}^6 \to \mathbb{R}^6$ in **Voigt space**.

**Redistribution within each microplane.** In Eqs. 5 and 6, we first redistribute stresses **within each microplane**. We group the components belonging to the same plane as simple 3-vectors, for example

$$\boldsymbol{\sigma}^x = \begin{bmatrix} \sigma_{xx} \\ \sigma_{xy} \\ \sigma_{xz} \end{bmatrix}, \quad \boldsymbol{\sigma}^y = \begin{bmatrix} \sigma_{yy} \\ \sigma_{yz} \\ \sigma_{yx} \end{bmatrix}, \quad \boldsymbol{\sigma}^z = \begin{bmatrix} \sigma_{zz} \\ \sigma_{zx} \\ \sigma_{zy} \end{bmatrix}. \tag{22}$$

On each plane, we apply a learned $3 \times 3$ linear map:

$$\tilde{\boldsymbol{\sigma}}^x = (\mathbf{I}_3 + C_x)\, \boldsymbol{\sigma}^x; \tilde{\boldsymbol{\sigma}}^y = (\mathbf{I}_3 + C_y)\, \boldsymbol{\sigma}^y; \tilde{\boldsymbol{\sigma}}^z = (\mathbf{I}_3 + C_z)\, \boldsymbol{\sigma}^z, \tag{23}$$

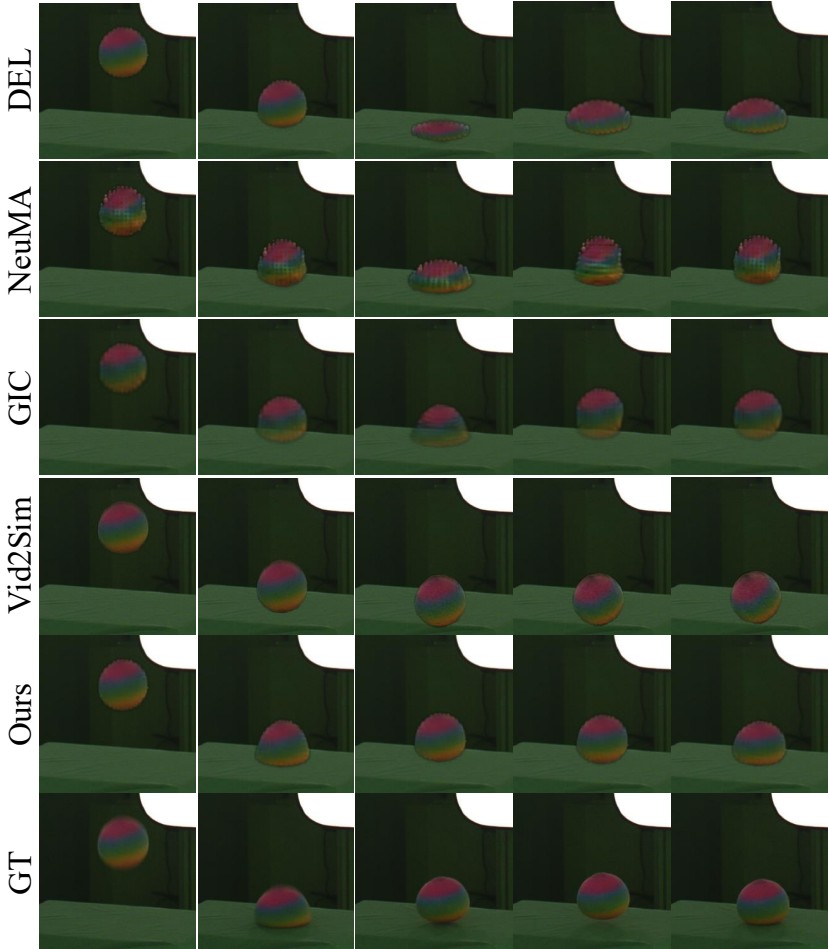

*Figure 18.* Qualitative comparison of different baselines and our methods on the Rainbowball scenario.

which is exactly what Eq. 5 does. Each entry of $\tilde{\boldsymbol{\sigma}}^x$, for example, is just a linear combination of $(\sigma_{xx}, \sigma_{xy}, \sigma_{xz})$. Stacking all intermediate components back into a Voigt vector $\boldsymbol{\sigma}^{(1)}$ (and using only index reordering), this whole "within-plane" redistribution is again a $6 \times 6$ matrix acting on $\boldsymbol{\sigma}$:

$$\boldsymbol{\sigma}^{(1)} = \mathbf{B}_{\mathrm{plane}}\, \boldsymbol{\sigma}, \qquad \mathbf{B}_{\mathrm{plane}} \in \mathbb{R}^{6 \times 6}. \tag{24}$$

Thus Eqs. 5-6 already define a valid linear map in Voigt form.

**Redistribution between microplanes.** Eq. 7 then mixes only the *normal* stresses from different planes, while keeping the shear components unchanged. Writing

$$\mathbf{n}^{(1)} = \begin{bmatrix} \sigma_{xx}^{(1)} \\ \sigma_{yy}^{(1)} \\ \sigma_{zz}^{(1)} \end{bmatrix}, \qquad \hat{\mathbf{n}} = \begin{bmatrix} \hat{\sigma}_{xx} \\ \hat{\sigma}_{yy} \\ \hat{\sigma}_{zz} \end{bmatrix}, \tag{25}$$

Eq. 7 can be written as

$$\hat{\mathbf{n}} = (\mathbf{I}_3 + C_{xyz})\, \mathbf{n}^{(1)}, \tag{26}$$

which is again a simple $3 \times 3$ linear map. The shear components are just copied over: $\hat{\sigma}_{yz} = \sigma_{yz}^{(1)}$, $\hat{\sigma}_{zx} = \sigma_{zx}^{(1)}$, $\hat{\sigma}_{xy} = \sigma_{xy}^{(1)}$. We deliberately apply cross-plane redistribution only to the normal stresses but not to the shear components. Normal stresses $(\sigma_{xx}, \sigma_{yy}, \sigma_{zz})$ control volumetric response and opening/closure of microplanes, and their interaction across directions (e.g., through Poisson-like effects) is physically meaningful. In contrast, each shear component (e.g., $\sigma_{xy}, \sigma_{yz}, \sigma_{zx}$) represents

sliding on a specific microplane and is already fully mixed within that plane by the per-plane maps $\mathbf{C}_x, \mathbf{C}_y, \mathbf{C}_z$ and the subsequent shear symmetrization. Introducing an additional cross-plane mixing layer for the shears would couple fundamentally different sliding modes (e.g., mixing $\sigma_{xy}$ with $\sigma_{yz}$), which is hard to justify physically and mainly increases the number of parameters without a clear benefit. Therefore, we restrict $C_{xyz}$ to act only on the normal block in Voigt space, leading to a clean block structure of the global $6 \times 6$ redistribution matrix.

Putting this together in Voigt form gives another $6 \times 6$ matrix $\mathbf{B}_{\text{norm}}$ such that

$$\hat{\boldsymbol{\sigma}} = \mathbf{B}_{\text{norm}} \, \boldsymbol{\sigma}^{(1)}, \qquad \mathbf{B}_{\text{norm}} \in \mathbb{R}^{6 \times 6}, \tag{27}$$

where the upper $3 \times 3$ block of $\mathbf{B}_{\text{norm}}$ is $\mathbf{I}_3 + C_{xyz}$ and the lower block is the identity.

**Putting everything together and the global redistribution matrix.** Combining the "within-plane" redistribution Eq. 24, the fixed shear symmetrization (a matrix $\mathbf{S}$ with entries $0, 1, \frac{1}{2}$), and the "between-plane" redistribution Eq. 27, we obtain the overall Voigt-form multiplication

$$\hat{\boldsymbol{\sigma}} = \mathbf{B}_{\text{norm}} \, \mathbf{S} \, \mathbf{B}_{\text{plane}} \, \boldsymbol{\sigma} \; = \; \mathbf{A} \, \boldsymbol{\sigma} \in \mathbb{R}^{6 \times 6}. \tag{28}$$

Equivalently, we can write this global redistribution explicitly as

$$\begin{bmatrix} \hat{\sigma}_{xx} \\ \hat{\sigma}_{yy} \\ \hat{\sigma}_{zz} \\ \hat{\sigma}_{yz} \\ \hat{\sigma}_{zx} \\ \hat{\sigma}_{xy} \end{bmatrix} = \mathbf{A}^{6 \times 6} \begin{bmatrix} \sigma_{xx} \\ \sigma_{yy} \\ \sigma_{zz} \\ \sigma_{yz} \\ \sigma_{zx} \\ \sigma_{xy} \end{bmatrix}. \tag{29}$$

Each entry $a_{\alpha\beta}$ is the final coefficient of $\sigma_\beta$ in the linear combination that produces $\hat{\sigma}_\alpha$; It is obtained by "filling" contributions from the within-plane maps $C_x, C_y, C_z$ and the between-plane map $C_{xyz}$ through the product $\mathbf{B}_{\text{norm}} \mathbf{S} \mathbf{B}_{\text{plane}}$.

The key point is simple: $\mathbf{B}_{\text{plane}}$, $\mathbf{S}$, and $\mathbf{B}_{\text{norm}}$ are all linear operators, and their product is still a linear operator $\mathbf{A}$. Therefore, there exists a $\mathbf{C}$ that satisfies

$$\mathbf{A} = \mathbf{I}_6 + \mathbf{C}, \quad \Rightarrow \quad \hat{\boldsymbol{\sigma}} = (\mathbf{I}_6 + \mathbf{C}) \, \boldsymbol{\sigma}, \tag{30}$$

which has exactly the same form as (21) and thus corresponds to an equivalent fourth-order tensor $\mathbb{C}_{ijkl}$.

Intuitively, we do not change the fact that each new stress component is a linear combination of all old stress components. We simply factor the large matrix $(\mathbf{I}_{6 \times 6} + \mathbf{C})$ into three steps: first redistributing stresses *within* each microplane (controlled by $C_x, C_y, C_z$), then symmetrizing shear pairs, and finally redistributing the normal components *across* microplanes (controlled by $C_{xyz}$). This keeps the mapping linear and physically reasonable, while turning the otherwise opaque fourth-order tensor $C_{ijkl}$ into a sequence of more interpretable operations.

## P. Constitutive Models as Physical Priors

A constitutive model explains how a material reacts to stress, strain, or other external forces. It defines the material's behavior by connecting stress and strain through equations, which can describe complex behaviors like elasticity, plasticity, and fracture. The MPM simulation can model a wide range of materials by using different constitutive models.

Our approach also relies on a constitutive model as a prior, which is then used to perform adaptive redistribution and correction of the stress based on this model. For any method requiring a constitutive model, we select the appropriate model from those listed in this chapter using a large language model, and it remains the same across different models. Generally, Cauchy stress $\sigma$ is represented as

$$\sigma = \frac{1}{J} \frac{\partial \Psi}{\partial F}(F^E) F^{E^T} \tag{31}$$

in which $J = \det(F)$, $\Psi$ is the energy density function to describe the behavior of a certain material, which can be seen as a function of the elastic component of $F$ ($F^E$). The $F^E$ is projected by $F^E = \psi(F)$, which is the so-called return mapping function. Both $\psi$ and $\Psi$ are referred to as constitutive models. We define the Cauchy stress $\sigma$ for these models as follows:

**Linear Cauchy Stress**. The simplest practical constitutive model is linear elasticity, defined in terms of the small strain tensor, or the infinitesimal strain tensor. This strain tensor will give rise to a computationally lightweight constitutive model.

$$J\sigma = \mu(F + F^T - 2I)F^T + \lambda\operatorname{tr}(F - I)F^T \tag{32}$$

**Neo-Hookean Cauchy Stress**. NeoHookean elastic model is a widely used nonlinear hyperelastic framework, for simulating material elasticity and predicting deformations, which can be expressed as:

$$J\sigma = \mu\left(FF^T - I\right) + \lambda\log(J)I \tag{33}$$

**Fixed Corotated Cauchy Stress**. Another simple and widely used model that is defined from the Singular Value Decomposition (SVD) is the so-called fixed corotated model.

$$J\sigma = 2\mu\left(F - R\right)F^T + \lambda(J - 1)JI \tag{34}$$

where $R = UV^T$ and $F = U\Lambda V^T$, which represent the singular value decomposition of the elastic deformation gradient. $J$ is the determinant of $F$.

**St. VK Cauchy Stress**. St. Venant-Kirchhoff model offers significant benefits over a linear elastic model, which is a rotationally invariant model.

$$J\sigma = U\left(2\mu\epsilon + \lambda\operatorname{sum}(\epsilon)1\right)V^T F^T \tag{35}$$

in which $\epsilon = log(\Lambda)$. Other symbols remain the same meaning with previously defined.

In our study, the deformation gradient is multiplicatively decomposed as $F = F^E F^p$, based on a yield stress condition in all plasticity models. We also can obtain the plastic projection process as $F^E = \psi(F)$. Then a hyperelastic constitutive model is applied to compute the Cauchy stress $\sigma$. For a purely elastic continuum, we set $F^E = F = \psi(F)$.

**Drucker-Prager Mapping**. The return mapping of Drucker-Prager plasticity can be defined as :

$$F^E = UZ(\Sigma)V^T \tag{36}$$

$$Z(\Sigma) = \begin{cases} 1, & \text{if } \sum(\epsilon) > 0, \\ \Sigma, & \text{if } \delta\gamma \le 0, \sum(\epsilon) \le 0, \\ \exp\left(\epsilon - \delta\gamma\frac{\hat{\epsilon}}{\|\hat{\epsilon}\|}\right), & \text{otherwise.} \end{cases} \tag{37}$$

where $\delta\gamma = \|\hat{\epsilon}\| + \alpha\left(\frac{(d\lambda+2\mu)\sum(\epsilon)}{2\mu}\right)$, $\alpha = \sqrt{\frac{2}{3} \cdot \frac{2\sin\varphi_f}{3-\sin\varphi_f}}$, and $\phi_f$ is the friction angle.

**Von Mises Mapping**. Von Mises shares the same mapping function as the Drucker-Prager (36).

$$Z(\Sigma) = \begin{cases} \Sigma, & \delta\gamma \le 0, \\ \exp\left(\epsilon - \delta\gamma\frac{\hat{\epsilon}}{\|\hat{\epsilon}\|}\right), & \text{otherwise.} \end{cases} \tag{38}$$

Here $\delta\gamma = \|\hat{\epsilon}\|_F - \frac{\tau_Y}{2\mu}$, $\tau_Y$ is the yield stress.

**Herschel-Bulkley Mapping**. We implement Continuum foam and we follow the simple version of that described in PhysGaussian, $s$ can be recovered as $\mathbf{s} = \mathbf{s} \cdot \frac{\mathbf{s}^{\text{trial}}}{\|\mathbf{s}^{\text{trial}}\|}$,

$$s = s^{\text{trial}} - \left(s^{\text{trial}} - \sqrt{\frac{2}{3}}\sigma_Y\right) / \left(1 + \frac{\eta}{2\mu\Delta t}\right) \tag{39}$$

The corresponding Kirchhoff stress can be:

$$\tau = \frac{\kappa}{2}\left(J^2 - 1\right)\mathbf{I} + \mu\operatorname{dev}\left[\det(\mathbf{b}^E)^{-\frac{1}{3}}\mathbf{b}^E\right] \tag{40}$$

in which $\mathbf{b}^E = \mathbf{F}^E\mathbf{F}^{E^T}$

## Q. Overview of Material Point Method

The Material Point Method (MPM) is a numerical simulation method for solving continuum mechanics problems. MPM uses particles and a background grid as discrete elements of the simulation domain, enabling the modeling of a wide range of materials. It solves problems by transferring mass and momentum between the particles and the background grids during simulation and performing computations on the grid. It is gonverned by the momentum and mass conservation in the Eulerian form:

$$\frac{d\rho}{dt} = -\rho\nabla \cdot \mathbf{v}, \ \rho\mathbf{a} = \nabla \cdot \sigma + \rho\mathbf{b}. \tag{41}$$

where $\rho$ denotes the density, $v$ refers to the velocity, $\mathbf{a}$ is the acceleration. $\sigma$ represents the Cauchy stress tensor and $\mathbf{b}$ corresponds to body forces. Mass conservation is naturally preserved in MPM by transporting the Lagrangian particles.

In practical MPM implementations, the integrals are computed using particle-based quadrature, where each material point contributes to the integrals based on its mass and position. The test functions $w$ are typically chosen from a set of shape functions defined on the computational background mesh.

**Spatial Discretization** To numerically solve the weak form derived earlier, the Material Point Method (MPM) performs spatial discretization by leveraging shape functions defined over the computational background grid. These shape functions serve to transfer information between the Lagrangian material points and the Eulerian background mesh, thus enabling a hybrid discretization approach.

Specifically, the test functions $w$ and the acceleration field $a$ are approximated using a set of basis functions $\{N_a\}$ defined on the grid nodes. This results in the following semi-discrete equation for each grid node $a$:

$$\sum_{b=1}^{N_G} M_{ab}a(b) = -\sum_{i=1}^{N_P} V_0(i)\boldsymbol{\tau}(i)\nabla N_a(x(i))+$$
$$\sum_{i=1}^{N_P} M(i)N_a(x(i))b(i), \tag{42}$$

where $N_G$ is the total number of grid nodes and $N_P$ is the number of material points. The left-hand side of the equation involves the mass matrix $M_{ab}$, which accounts for the contribution of particle mass to the grid system. It is defined as:

$$M_{ab} = \sum_{i=1}^{N_P} M(i) \cdot N_a(x(i)) \cdot N_b(x(i)), \tag{43}$$

where $M(i)$ is the mass of particle $i$, and $N_a$, $N_b$ are the grid basis functions evaluated at the current position $x(i)$ of the material point. The discretization scheme effectively transfers the continuous balance of momentum equation into a system of equations posed on the grid. MPM uses this system to compute the nodal accelerations, which are then interpolated back to the particles during the update step.

**Temporal Discretization** To advance the solution in time, the MPM typically employs an explicit time integration scheme due to its simplicity and ease of implementation. In this formulation, we adopt the *explicit forward Euler method*, which approximates time derivatives using finite differences over a discrete time step $\Delta t$.

Starting from the semi-discrete system obtained from spatial discretization, the temporal discretization updates the nodal velocity $v(b)$ at each grid node $b$ from time $t$ to $t + 1$ as follows:

$$\sum_{b=1}^{N_G} M_{ab}\frac{v_{t+1}(b) - v_t(b)}{\Delta t} = -\sum_{i=1}^{N_P} V_0(i)\boldsymbol{\tau}_t(i)\nabla N_a(x_t(i))$$
$$+ \sum_{i=1}^{N_P} M(i)N_a(x_t(i))b_t(i), \tag{44}$$

In practice, the mass matrix $M_{ab}$ is often diagonalized (lumped) to decouple the equations for each node, allowing for a simple explicit velocity update:

$$v_{t+1}(a) = v_t(a) + \frac{\Delta t}{M_{aa}}f_{\text{net}}(a),$$

where $f_{\text{net}}(a)$ includes both internal and external force contributions. The explicit Euler scheme is conditionally stable, requiring small enough $\Delta t$ to satisfy a CFL-like condition that depends on wave speed and grid spacing.

