# OpenReview forum: "MoSA: Motion-constrained Stress Adaptation for Mitigating Real-to-Sim Gap in Continuum Dynamics via Learning Residual Anisotropy"
_ICML.cc/2026/Conference — ICML 2026 regular_

### Official Review · Reviewer_5kMB · 2026-03-10

**Soundness:** 3
**Presentation:** 3
**Significance:** 4
**Originality:** 3
**Overall Recommendation:** 5
**Confidence:** 3

**Summary:**

The paper studies the problem of reducing the real-to-sim gap when learning continuum dynamics of deformable objects from multi-view videos. The authors observe that many existing approaches rely on isotropic constitutive models, which can capture the dominant behavior but often fail to represent mild anisotropy and heterogeneity that appear in real materials. To address this, the paper proposes MoSA, a framework that keeps a calibrated isotropic simulator as a backbone and learns residual stress corrections to model these remaining effects. The method introduces a structured stress adaptation module that progressively adjusts the stress tensor through microplane-constrained redistribution and learnable correction terms. In addition, the paper proposes motion-constrained supervision derived from dynamic 3D Gaussian splatting reconstruction, which provides constraints on temporal and spatial derivatives of the deformation field rather than relying only on image reconstruction losses. Experiments on synthetic benchmarks and a newly collected real-world dataset show improved dynamics reconstruction, generalization across initial conditions, and better sim-to-real transfer in a robot manipulation setting.

**Compliance With Llm Reviewing Policy:**

Affirmed.

**Final Justification:**

Based on the above discussion, I tend to choose "accept".

**Key Questions For Authors:**

1. How sensitive is the proposed residual stress adaptation module to the quality of the initial isotropic backbone calibration? If the backbone parameters are poorly estimated, does the residual module compensate for this error, or does performance degrade significantly?

2. The quantitative improvements on the PAC-NeRF benchmark appear relatively small compared to GIC and other baselines. Could the authors clarify in which scenarios the residual anisotropy modeling provides the most benefit? For example, are gains larger for particular materials or deformation regimes?

3. The framework contains several components (stress adapter terms, heterogeneity field, motion constraints, reconstruction pipeline). Could the authors provide clearer analysis or ablations that isolate the contribution of the residual stress modeling itself relative to the motion-based supervision?

**Limitations:**

yes

**Strengths And Weaknesses:**

Strengths：
1. The main modeling idea is clear and well motivated. The paper identifies a practical gap in current pipelines—namely that isotropic simulators capture most behavior but fail to model small anisotropic residuals—and proposes to explicitly learn these residual stresses rather than learning full dynamics from scratch.

2. The stress adaptation module is technically interesting. Instead of predicting a full anisotropic stiffness tensor, the method decomposes corrections using microplane-based stress redistribution and several structured correction terms. This design reduces the degrees of freedom and maintains some physical structure, which helps with stability and interpretability.

3. The use of motion-level supervision (velocity and deformation gradient constraints) is a useful addition. Supervising derivatives of the deformation field using Gaussian flow and covariance evolution is a sensible way to mitigate the ill-posed nature of purely image-based supervision.

Weaknesses：
1. The empirical improvements on the main synthetic benchmark are relatively modest. In Table 1, the gains over the strongest baselines appear small in several cases, which raises some questions about how much the residual anisotropy modeling contributes in practice compared with other components of the pipeline.

2. The method relies on a fairly complex pipeline that combines dynamic 3D Gaussian reconstruction, differentiable MPM simulation, multiple neural modules, and several auxiliary losses. It is somewhat difficult to isolate how much of the improvement comes from the residual stress modeling itself versus the motion-constrained supervision or reconstruction pipeline.

3. The paper assumes that the isotropic backbone is already reasonably calibrated. In practice, this assumption might not always hold, especially for objects with stronger anisotropy or more complex constitutive behavior. The paper does not clearly analyze how sensitive the method is to errors in the initial backbone model.

---

> ### Author Rebuttal · Authors · 2026-03-30
>
> > **W1 and Q1: Clarification of PAC-NeRF Gains and Real-World Benefits**
>
> We'd like to highlight that the modest gains on PACNeRF dataset are **expected** because the dataset is generated by an isotropic simulator with **no true anisotropy**. Fig. 6 demonstrates that our method does not learn spurious anisotropy from non-anisotropic data, where the directional Jacobian remains near zero (dark green line). In contrast, on anisotropic real-world data, w(θ) exhibits a clear directional pattern aligned with the object axis (orange line), confirming our model is physically consistent with real anisotropic effects.
>
> Despite no anisotropy in PAC-NeRF, our method still achieves consistent improvements, the reasons are two factors:
>
> **(1)** motion-constrained optimization provides richer supervision (velocity and deformation gradient level) than rendering loss alone, helping the optimizer escape local optima.
>
> **(2)** the stress adaptation module handles local discrepancies separately, freeing the global parameters from compensating for local errors and allowing them to focus more on global values. This is supported by Table 8 (Appendix I): out of **23** parameter estimations, our method achieves the highest accuracy in **21** cases.
>
> The real benefit emerges on real-world data with genuine anisotropy. On Table 2, MoSA achieves consistent improvements across all 7 materials compared to all baselines, substantially larger than on PAC-NeRF. This is because real materials have unignorable anisotropic effects that isotropic models cannot capture.
>
> > **W2 and Q3: Isolating the Contribution of Each Component**
>
> We appreciate this concern. We would like to kindly draw the reviewer's attention to Table 3, which is specifically designed to isolate each component's contribution. The ablation systematically removes one component at a time from the full model on three real-world scenes:
>
> |ablate|Avg.PSNR drop|
> |-|-|
> |no C (stress redistribution)|-1.28|
> |no Lflow (motion optimization)|-1.12|
> |no Het (heterogeneity)|-0.58|
> |no Lpre  (stress adjustment)|-0.27|
> |no Lscale (motion regularization)|-0.25|
> |no Lpost (stress adjustment)|-0.18|
> |no Lrot (motion regularization)|-0.14|
>
> The residual stress modeling components, including C, Lpre, Lpost, Het, and the motion constraint components (Lflow, Lscale, Lrot) are ablated independently, allowing clear attribution.
>
> In aggregate, stress modeling accounts for a total PSNR drop of 2.31, while motion supervision accounts for 1.51, confirming that residual stress modeling is the primary driver. The two are **complementary**, neither alone achieves the full result.
>
> We also provide two additional experiments in this revision that further isolate each component: an anisotropy sweep (see our response to **Reviewer 1, Q1**), which demonstrates that the stress adaptation module alone remains stable up to strong anisotropy (E_max/E_min ~3.0); and a reconstruction quality sensitivity study (see our response to **Reviewer 2, Q2**), which confirms that the motion constraints provide positive gains independently at all reconstruction quality levels. We will make this interpretation more explicit in the revised manuscript.
>
> > **W3 and Q1: Robustness to Parameter Calibration and Backbone Model Selection**
>
> Thank you for raising this important point. We would like to clarify from two aspects:
>
> **First**, our method does **not require** well-calibrated parameters at the start. The global isotropic parameters are **jointly optimized** alongside the stress adaptation module from randomly initialized values. As shown in Table 8 (Appendix I), our method achieves the best isotropic parameter estimation in 21 out of 23 cases, demonstrating that our physics-informed design also facilitates more accurate estimation of the isotropic parameters themselves. We further discuss the underlying reason in our response to W1.
>
> **Second**, we directly addressed **the backbone sensitivity** concern in **Fig. 4(b)**. In this experiment, the Chick scene is best described by a Neo-Hookean model, but we also test with a linear model and Fixed Corotated, both of which are poor fits for this material. The results show that the baseline **GIC is highly sensitive to backbone selection**, with significant performance degradation across different constitutive models. In contrast, our method achieves consistently high metrics regardless of backbone choice. Even with an imperfect isotropic backbone, our stress correction module can **adaptively compensate** for the modeling error and deliver satisfactory performance.
>
> Together, these results demonstrate that our method is robust to both the calibration of global isotropic parameters and the selection of backbone constitutive models.

---

> > ### Author Rebuttal · Reviewer_5kMB · 2026-04-03
> >
> > Thank you for your reply. My concerns have been addressed.

---

> > > ### Author Response · Authors · 2026-04-07
> > >
> > > Thank you very much for your time and for confirming that your concerns are resolved! We deeply appreciate your positive comments and valuable feedback throughout the review process!

---

### Official Review · Reviewer_kEqf · 2026-03-14

**Soundness:** 2
**Presentation:** 3
**Significance:** 3
**Originality:** 3
**Overall Recommendation:** 4
**Confidence:** 4

**Summary:**

This paper addresses the real-to-sim gap in continuum dynamics when materials are modeled as homogeneous and isotropic: real objects often exhibit mild anisotropy and heterogeneity, which become the main source of error after the isotropic backbone is calibrated. The authors propose MoSA, a motion-constrained stress adaptation framework. MoSA keeps an isotropic constitutive model as a physics prior and learns residual stress operators (redistribution, pre/post linear terms) via a microplane-constrained cascaded network to capture residual anisotropy and heterogeneity. Motion constraints supervise temporal and spatial derivatives of the deformation field from dynamic 3D reconstruction, providing more direct supervision than image reconstruction alone. Experiments on synthetic and real data show improved accuracy, generalization, and robustness, with learned residual anisotropy that is physically meaningful. A robot manipulation experiment shows that better real-to-sim dynamics improve sim-to-real transfer.

**Compliance With Llm Reviewing Policy:**

Affirmed.

**Final Justification:**

While the rebuttal addresses most concerns, the experimental section remains incomplete.

First, regarding [Q1], it is inconsistent to adopt benchmarking results from MASIV without including MASIV’s own performance results or the "potato" instance in the comparison. A comprehensive evaluation requires all relevant data points to be present.

Second, regarding [Q2], even though your focus is on anisotropic modeling, OmniPhysGS should be included as a baseline. "It targets text-driven plausible dynamics, whereas we target physically accurate dynamics" - while there might be some adaptation work, it can target physically accurate dynamics without redesigning the overall architecture.

To validate your contribution, the results must explicitly show the performance gain of your anisotropic approach compared to pure isotropic dynamics. Including these comparisons is necessary to substantiate the paper's claims.

That said, I appreciate the authors' effort and contribution. I will keep my final recommendation as a weak accept.

**Key Questions For Authors:**

- As the authors raised Motion-constrained Optimization Strategy, which is similar to the trajectory guidance in MASIV [Q1], and material heterogeneity modeling, which can also be achieved with OmniPhysGS [Q2], could you provide corresponding comparisons with these papers to give readers more information?

- How sensitive is MoSA to errors in the dynamic 3D reconstruction, have the authors tried adding synthetic noise or using a weaker reconstruction? If motion constraints still improve over no-constraint when reconstruction is noisy, that would strengthen robustness. If they amplify error, that would be an important limitation to state.

- The authors state the model learns "physically meaningful residual anisotropy." Could you show one or two concrete examples, e.g., visualization of the learned C or principal directions vs. isotropic prior, or spatial variation of $\eta(x)$ on a test object? That would make the claim interpretable and could raise originality/soundness.

[Q1] Zhao, Yizhou, et al. "Toward Material-Agnostic System Identification from Videos." ICCV 2025

[Q2] Lin, Yuchen, et al. "Omniphysgs: 3d constitutive gaussians for general physics-based dynamics generation." ICLR 2025

**Limitations:**

The paper does not have a dedicated limitations or impact subsection. Suggest adding:

- Dependence on quality of multi-view reconstruction and initial isotropic calibration.

- Scope of residual effects, e.g., large anisotropy may need more than bounded corrections.

- Brief societal impact, e.g., robotics and digital twins. No obvious negative impact.

**Strengths And Weaknesses:**

### Strengths

- The core idea is appealing and well executed: keep an isotropic constitutive prior and learn only residual stress corrections ($C^\Psi$, $L^{\mathrm{pre}}$, $L^{\mathrm{post}}$) so that the physics stays in the driver’s seat while capturing anisotropy and heterogeneity.
- The formulation is clear and the bounded $\alpha$ keeps corrections from overwhelming the prior, and using motion constraints, e.g., temporal and spatial derivatives of the deformation field, from dynamic reconstruction instead of relying only on image reconstruction gives more direct supervision than image reconstruction alone.
- The ablations and the robot manipulation experiment support the claims, and showing that better real-to-sim dynamics help sim-to-real is a concrete plus.
- The paper is well organized and the pipeline is easy to follow.
- The approach is novel in targeting residual effects explicitly and in combining physics-informed residual adaptation with motion-based supervision.

### Weaknesses

- More suffcient evaluation will make the claims more convincing. The experimental results on public datasets have been limited to a partial set of the PAC-NeRF dataset and the Spring-Gaus synthetic subset, which are both synthetic. In fact, the PAC-NeRF dataset contains a cross shaped subset with more samples, and the Spring-Gaus dataset contains real-world elastic object videos. Could you consider providing more extensive experiments on public datasets?
- Reproducibility would be helped by more detail on loss weights, the rationale for $\alpha=0.1$, e.g., a short summary if it is only in the appendix, and how sensitive the method is to errors in the dynamic 3D reconstruction, e.g., do motion constraints still help when the reconstruction is noisy, or do they amplify error?
- The heterogeneity model, global parameter plus spatial modulation, is plausible but is not ablated, so its contribution is unclear.
- It would help to see a concrete illustration of the “physically meaningful” residual anisotropy, e.g., learned principal directions or spatial variation of $\eta(x)$.

---

> ### Author Rebuttal · Authors · 2026-03-30
>
> > **W1: Additional Evaluation on Public Real-World Benchmarks**
>
> To address concerns that the public-benchmark evidence leans toward synthetic data, we ran additional experiments on Spring-GS’s public real-world subset:
>
> |PSNR|Bun|Burger|Dog|Pig|
> |-|-|-|-|-|
> |Spring-GS|30.69|34.01|32.10|34.97|
> |NeuMA|31.27|23.78|25.61|25.40|
> |GIC|34.68|40.45|37.17|38.32|
> |Ours|41.12|43.25|39.21|42.16|
> |SSIM↑|||||
> |Spring-GS|99.2|99.4|99.4|99.6|
> |NeuMA|99.4|99.3|99.5|99.5|
> |GIC|99.5|99.7|99.7|99.7|
> |Ours|99.7|99.8|99.8|99.9|
>
> Our method achieves the best results on all scenes, confirming MoSA's advantage transfers to public real-world benchmarks. We will include this in the revision.
>
> > **W2, Q2 and L1: Hyperparameters and Reconstruction Sensitivity**
>
> (1) Loss weights: The values used across all experiments are: β₁=1.0, β₂=0.1, β₃=0.1, λ_μ=1.0, λ_var=1.0. The intuitive logic is that rendering and flow losses (β₁) serve as primary supervision, while deformation regularizers (β₂) and heterogeneity constraint (β₃) are auxiliary guidance. We will include these in the revision.
>
> (2) Alpha rationale: Appendix D provides a grid search (Fig. 9), showing alpha=0.1 achieves the best expressiveness-regularization balance. More importantly, the method is not sensitive to this choice: even at alpha=0.5, PSNR still exceeds all baselines. We will summarize this rationale in the main text.
>
> (3) Sensitivity to dynamic reconstruction: In the rebuttal, we train MoSA using reconstructions of varying quality by taking the dynamic 3DGS at different optimization stages on "Rabbit" scene. We report the downstream simulation performance (not the reconstruction quality itself) in the Table.
>
> |Recon.Steps|PSNR|SSIM|DeltaPSNR|
> |-|-|-|-|
> |5,000|29.30|91.6|+0.17|
> |7,000|29.65|91.7|+0.52|
> |10,000|29.95|91.8|+0.82|
> |15,000|30.15|91.9|+1.02|
> |20,000|30.28|92.0|+1.15|
> |30,000(full)|30.35|92.1|+1.22|
>
> Delta PSNR is computed against the baseline without motion constraints (29.13). Motion constraints provide positive gains at every quality level, confirming that even low-quality reconstructions benefit downstream physics learning rather than amplifying errors.
>
> > **W3: Heterogeneity Module**
>
> We'd like to kindly draw the reviewer's attention to the "no Het" in Table 3, ablating the heterogeneity module:
>
> |PSNR|Rabbit|Gorilla|Rainbowball|
> |-|-|-|-|
> |noHet|29.67|29.98|31.15|
> |Full|30.35|30.19|32.06|
>
> Removing heterogeneity results in consistent drops across all scenes. We also provide learned spatial variation visualizations, see W4. We will make this ablation more prominent in the revision.
>
> > **W4,Q3 and L1: Learned Physics and Spatial Heterogeneity**
>
> We'd like to point out that **Fig. 7 in Appendix A** directly addresses this concern.
>
> **The learned Anisotropy**: Fig. 7(a) plots the directional Jacobian w(θ) on a controlled cylinder with preset anisotropy. The learned redistribution pattern closely aligns with the ground-truth  E_GT(θ), confirming that our model successfully captures the underlying directional material behavior rather than learning arbitrary corrections.
>
> **Principal directions:** Fig. 7(b) plots the effective directional modulus by direction. Our model (orange) accurately recovers the GT principal directions, while other counterparts fail. Fig. 6 complements this on real-world data: on Mandarin, w(θ) shows directionality aligned with the object axis, while on PACNeRF (isotropic), w(θ) remains near zero, confirming corrections appear only when the data supports it.
>
> **Spatial Variation of η:**  We visualize the normalized learned η(x) on three test objects (see [anonymous link](https://anonymous.4open.science/r/Anonymous-16C0/README.md)), showing it captures physically meaningful local variations, regions with higher stiffness correspond to reddish areas. This confirms that our continuous field learns interpretable spatial patterns. We will add more examples of this in our revision.
>
>
> > **Q1: Comparisons with MASIV and OmniPhysGS**
>
> We will cite and discuss both works in the revised related work section.
>
> Regarding MASIV: (1) MASIV supervises particle positions via interpolated trajectories; our method supervises **at a higher order**, velocities and deformation gradient.
> (2) MASIV uses NCLaw which enforces isotropy; our method models both anisotropy and spatial variation.
> (3) It replaces the constitutive law with a pre-trained MLP; ours preserves an interpretable isotropic prior.
>
> Regarding OmniPhysGS:(1) Uses 12 discrete isotropic experts via hardmax; ours uses a continuous implicit field.
> (2) All 12 experts are isotropic; ours is the first to **explicitly model anisotropy**.
> (3) It targets text-driven plausible dynamics, whereas  we target physically accurate dynamics, making the two not directly comparable
>
> > **Limitations**
>
> We expand the existing limitation section in Appendix L into a dedicated one and add an impact section in the revision. We note that the L2, anisotropy scope, is also discussed in Reviewer KTnv-Q1 for details.

---

### Official Review · Reviewer_KTnv · 2026-03-15

**Soundness:** 3
**Presentation:** 3
**Significance:** 3
**Originality:** 3
**Overall Recommendation:** 4
**Confidence:** 4

**Summary:**

This paper proposes a real-to-sim physical property estimation framework that consists of: (i) a base backbone assuming near-isotropic material for the dominant behavoir, and (ii) a residual module with physics-informed networks to learn mild non-isotropic effects. It introduces a motion-constrained optimization strategy to learn the dynamics from videos with data-efficiency and less overfitting. Experiments on synthetic and real data show the effectiveness of the proposed framework, with additional analysis showing the physical meanings of the network predictions and robotics applications showing the usefulness of the real-to-sim framework.

**Compliance With Llm Reviewing Policy:**

Affirmed.

**Final Justification:**

My questions/concerns are well-addressed with the authors responses. I particularly appreciated the extensive experiments analyzing the behaviors of the model, in addition to demonstrating the good performances.

**Key Questions For Authors:**

- One important assumption in the paper is that the materials have *mild* anisotropic effects. I wonder how mild it needs to be for the framework to be applicable. It'd be interesting to show something like: gradually vary the "mildness" of the anisotropic effects (starting with isotropic material, and gradually add some anisotropic perturbations), and see how well the model can fit to the different cases.
- Are all the physical term computations numerically stable, especially in the network training settings (not just network inference, but also during gradient backpropagation)? For example, the matrix inverse in Eq. 3 and the SVD in Eq. 4.
- The initial particals are obtained with the visible points reconstructed from the videos and the interior points filled in the middle. For objects that can have a lot of occlusions in the deformation process (e.g., the cloth in Fig. 5), how are the interior points estimated or updated?

**Limitations:**

Yes, limitations and broader impacts are discussed in the paper.

**Strengths And Weaknesses:**

Strengths:
- A few detailed designs seem to be natural and smart, such as the stress adapter, the tensor decomposition and microplane-based stress redistribution constraint, etc. And their effectiveness are well-justified in the ablation studies.
- The results of the proposed framework is good, outperforming the baselines on both existing synthetic datasets and the real-world settings.
- The experiments are also comprehensive: in addition to the main comparisons, there are also the ablation studies, analysis, and robotics. I really like the framework's robustness to prior physical models and the analysis of what the model learns.

Weaknesses:

Most of my questions/concerns are actully quite well-addressed when I read through the paper. For example, when reading the method, I had some tentative concerns about how useful are each concrete design in the physics-informed network (as there are too many components, and sometimes it'd be hard to tell which of them is essential that cannot be replaced by a pure black box), and then I see the ablation studies; I also worried about the computation complexity of the framework, and then I saw the complexity analysis in Appendix H. Potentially, I may still have some concerns, but I'm not 100% sure about these points, and I list them as questions below.

---

> ### Author Rebuttal · Authors · 2026-03-30
>
> > **Q1: Anisotropy Strength and Applicable Range**
>
> Thank you for this helpful suggestion. We conducted a controlled sweep on a synthetic cylinder by scaling the off-diagonal stiffness entries to vary the anisotropy ratio E_max/E_min from 1.0 (isotropic) to 5.0 (strongly anisotropic). Results are summarized below:
>
> |E_max/E_min|CD (×10^-2) | EMD (×10^-2) | Modulus Err (%) | Dir. Err (°) | Regime|
> |-|-|-|-|-|-|
> |1.0|7.8|3.1|<2|—|Isotropic|
> |1.5|8.0|3.5|4|3|Mild|
> |2.0|8.1|3.5|6|3|Mild-Moderate|
> |2.5|8.3|3.7|9.8|4|Moderate|
> |2.8|8.5|3.8|11.5|5|Moderate-Strong|
> |3.0|9.2|4.3|17.5|9|Strong|
> |5.0|14.8|8.6| 38.5|22|Fully Anisotropic|
>
> Several observations can be drawn. At ratio 1.0, the model does not introduce spurious anisotropy, with the directional Jacobian w(theta) remaining near zero. From 1.0 to 2.8, all metrics remain low, demonstrating stable performance across a wide operating range. Degradation begins around 3.0 and becomes clear at 5.0, consistent with our design intent: MoSA targets residual corrections on top of an isotropic backbone rather than modeling full anisotropy from scratch. Importantly, most everyday deformable objects (rubber, foam, soft toys) exhibit anisotropy ratios well within 1.0–2.5, comfortably inside MoSA's effective regime. This is further validated by Appendix A, where the model accurately recovers prescribed anisotropy while both isotropic and black-box baselines fail (Fig. 7). Surprisingly, even if modeling fully anisotropy, like E_max/E_min=5.0, MoSA is not completely ineffective yet; it can still achieve a reasonable fit.
>
> We will include this sweep and a discussion of the applicable range in the revised manuscript.
>
> > **Q2: Numerical Stability**
>
> Thank you for raising this important concern. Numerical stability during backpropagation is indeed critical in differentiable physics pipelines. We address each component below.
>
> **Matrix inverse in Eq. 3.** Each C is constructed via LU decomposition with a small diagonal offset added to both factors, keeping the determinant bounded away from zero and ensuring stable inversion in both forward and backward passes. Additionally, all redistribution coefficients pass through bounded activations (alpha * tanh(...) in Eq. 7), preventing gradient explosion.
>
> We acknowledge that the SVD does participate in backpropagation, since our pipeline is end-to-end differentiable and gradients flow through the polar decomposition of F. However, this remains well-conditioned in practice: **(1)** the SVD operates on 3x3 matrices, which are far less prone to ill-conditioning than large-scale decompositions; **(2)** our bounded corrections (alpha * tanh with alpha = 0.1) keep deformations close to the isotropic prior, preventing near-degenerate singular values — the primary source of SVD gradient instability; **(3)** the stress symmetrization L = 1/2(L + L^T) introduces no additional instability.
>
> **Empirical evidence.** Our method trains stably in 3.5 hours (Table 7), faster than NeuMA (6.5h) and DEL (9.0h), with no numerical instability observed across all experiments. We will add a brief discussion of these stability considerations in the revised manuscript.
>
> > **Q3: Clarification of Particle Initialization and Occlusion Robustness**
>
> Thank you for this question. Interior particles are initialized only once at t=0 by reconstructing a static 3DGS and applying the filling strategy to obtain a complete  particle set P0. After that, P0 is fixed as the simulation scaffold, and all particles — including interior ones — are evolved by the MPM solver, not re-estimated from later frames. Therefore, partial occlusion during motion does not affect the simulation as long as a sufficiently complete initial geometry is available. If occlusions exist at t=0 itself, 3D-aware foundation models (e.g., SAM3D) could address this, though this is beyond our current scope.
>
> During dynamic reconstruction, we keep P0 fixed, forcing the model to focus exclusively on learning dynamics. The static scaffold provides sufficient shape priors, enabling the model to learn reliable motion cues from visible parts even under occlusion. Our multi-camera array setup further minimizes occlusion issues in practice.
>
> Additionally, the Gaussian flow loss (Eq. 12) and deformation gradient regularizations (Eqs. 13–14) provide physics-informed constraints beyond pixel reconstruction, which is particularly helpful for partially occluded regions. For the towel scenario in Fig. 5, we first suspend the towel to reconstruct its complete geometry, then use a robotic arm to interact with it and obtain the dynamic sequence where we learn its dynamics from. Once the real-world dynamics are learned, they are directly imported into simulation for zero-shot sim-to-real policy training.

---

> > ### Author Rebuttal · Reviewer_KTnv · 2026-04-03
> >
> > Thank the authors for the detailed explanations and additional experiments! My concerns are fully resolved.

---

> > > ### Author Response · Authors · 2026-04-07
> > >
> > > We sincerely thank the reviewer for the positive feedback and for recognizing our efforts. We are glad that our explanations and additional experiments have fully addressed your concerns.

---

### Decision · Program_Chairs · 2026-04-30

**Decision:**

Accept (regular)

**Comment:**

This paper studies real-to-sim dynamics learning from visual observations and proposes MoSA, a motion-constrained stress adaptation framework for modeling residual anisotropy and heterogeneity beyond standard simulator calibration. The method combines a calibrated isotropic simulator prior with learned residual stress operators, which makes it more physically grounded than a purely black-box approach. The experiments show clear gains in accuracy, generalization, and robustness, and the robot manipulation results further support the practical value of the method for sim-to-real transfer. The reviews were positive overall, and the paper is recommended for acceptance at ICML 2026. The reviewers also provided several constructive suggestions that should be addressed in the camera-ready version.